# Genomic and Chemical Decryption of the Bacteroidetes Phylum for Its Potential to Biosynthesize Natural Products

Stephan Brinkmann,[a] Michael Kurz,[b] Maria A. Patras,[a] Christoph Hartwig,[a] Michael Marner,[a] Benedikt Leis,[a] André Billion,[a] Yolanda Kleiner,[a] Armin Bauer,[b] Luigi Toti,[b] Christoph Pöverlein,[b] Peter E. Hammann,[c] Andreas Vilcinskas,[a,d] Jens Glaeser,[a,c] Marius Spohn,[a] Till F. Schäberle[a,d]

[a]Fraunhofer Institute for Molecular Biology and Applied Ecology (IME), Branch for Bioresources, Giessen, Germany
[b]Sanofi-Aventis Deutschland GmbH, Frankfurt am Main, Germany
[c]Evotec International GmbH, Göttingen, Germany
[d]Institute for Insect Biotechnology, Justus-Liebig-University Giessen, Giessen, Germany

**ABSTRACT** With progress in genome sequencing and data sharing, 1,000s of bacterial genomes are publicly available. Genome mining—using bioinformatics tools in terms of biosynthetic gene cluster (BGC) identification, analysis, and rating—has become a key technology to explore the capabilities for natural product (NP) biosynthesis. Comprehensively, analyzing the genetic potential of the phylum Bacteroidetes revealed *Chitinophaga* as the most talented genus in terms of BGC abundance and diversity. Guided by the computational predictions, we conducted a metabolomics and bioactivity driven NP discovery program on 25 *Chitinophaga* strains. High numbers of strain-specific metabolite buckets confirmed the upfront predicted biosynthetic potential and revealed a tremendous uncharted chemical space. Mining this data set, we isolated the new iron chelating nonribosomally synthesized cyclic tetradeca- and pentadecalipodepsipeptide antibiotics chitinopeptins with activity against *Candida*, produced by *C. eiseniae* DSM 22224 and *C. flava* KCTC 62435, respectively.

**IMPORTANCE** The development of pipelines for anti-infectives to be applied in plant, animal, and human health management are dried up. However, the resistance development against compounds in use calls for new lead structures. To fill this gap and to enhance the probability of success for the discovery of new bioactive natural products, microbial taxa currently underinvestigated must be mined. This study investigates the potential within the bacterial phylum Bacteroidetes. A combination of omics-technologies revealed taxonomical hot spots for specialized metabolites. Genome- and metabolome-based analyses showed that the phylum covers a new chemical space compared with classic natural product producers. Members of the Bacteroidetes may thus present a promising bioresource for future screening and isolation campaigns.

**KEYWORDS** natural products, antifungal, NRPS, metabolomics, genomics, Bacteroidetes

The steady utilization of antimicrobials in all areas affected by human and animal diseases, but also in agriculture, supports the distribution of resistance across all environmental niches (1–4). Consequently, mortality rates accumulate—caused by multiresistant pathogens not treatable with approved anti-infectives (5). At the same time, approximately 15% of all crop production is lost to plant pathogen diseases nowadays (6). With devastating socioeconomic consequences for human health and food supply security, the increase of antifungal drug resistances has become a global problem (7). To sustain the medical care and the food supply for a growing population, continued innovation in discovery of new lead anti-infectives with improved activities and/or novel mechanisms of action is of greatest need (8, 9). Small molecules from biological origin—natural products

Address correspondence to Marius Spohn, marius.spohn@ime.fraunhofer.de, or Till F. Schäberle, till.f.schaeberle@agrar.uni-giessen.de.
The authors declare no conflict of interest.

(NPs)—are a rich source of new chemical entities and have always been a major inspiration for the development of anti-infectives and control agents. Between 1981 and 2019, 36.3% of new medicines based on small molecules and approved by the U.S. Food and Drug Administration (FDA) were NPs, derivatives of those or synthetic compounds that utilize an NP pharmacophore (10).

A promising NP group is represented by the cyclic lipopeptides (CLPs), sharing a common structural core composed of a lipid tail linked to a cyclized oligopeptide (11). CLP biosynthesis is phylogenetically dispersed over the bacterial kingdom. This translates into an immense structural diversity arising from differences in amino acid sequences (length, type, configuration, and modification) and the composition of the fatty acid moiety. Again, these variations result in heterogeneous biological activities of CLPs and promote their investigation in several research fields (11). For example, recently isolated isopedopeptins (12) overcome multidrug resistance of Gram-negative pathogens, while daptomycin (13) (Cubicin), an FDA-approved drug, is marketed for the treatment of complicated skin and soft tissue infections caused by Gram-positive bacteria (14). Today, further promising CLPs are already under clinical investigation and attack multiple molecular targets (11). In contrast, other CLPs like fengycin (15) exhibit an intrinsic antifungal activity (16, 17) and several CLP-producing bacteria have been registered with the U.S. Environmental Protection Agency (EPA) for their application as biocontrol agents (18). Their biosynthesis and production is mostly investigated in the genera *Pseudomonas*, *Bacillus*, and *Streptomyces* spp. (19). These taxa belong to the best-studied NP producers. Traditionally, antimicrobial NPs from bacterial origin had been isolated within large cultivation campaigns on those talented microorganisms. While those genera have delivered the vast majority of structurally different NPs over the last decades, they represent only a limited phylogenetic diversity (20). In addition to classical approaches, computational evaluation of genomic data paved the way to discover novel NPs in a target oriented manner by applying bioinformatics tools for biosynthetic gene cluster (BGC) identification, analyzation, and rating (21–26). These approaches provided evidence of a yet not fully exploited biosynthetic potential in terms of chemical novelty (27–31) certified with complementary metabolomics studies (32, 33). The predicted level of novelty increases by shifting from classical NP producing taxa toward a currently still underexplored phylogenetic and chemical space (34, 35). The validity of this dogma in NP discovery was recently shown, e.g., by the discovery of teixobactin (36) and darobactin (37), both produced by rare Proteobacteria.

The Bacteroidetes phylum represents such an underexplored phylogenetic space, too (34). Although, easy to cultivate and widely spread through all environments (38, 39), there are only a few biologically active molecules described from this taxonomic clade (e.g., isopedopeptins [12], elansolids [40], pinensins [41], formadicins [42], TAN-1057A-D [43], katanosins [44], and ariakemicins [45]). Whereas environmental studies already allowed a first glance at the genetic potential of the phylum (46), there is no systematic investigation of their actual genetic and metabolic repertoire and the overall number of isolated compounds remains low.

Motivated by this gap, we examined the NP biosynthesis potential of the Bacteroidetes phylum by computational analysis of 600 publicly available genomes for their BGC amount, type, and diversity. This revealed the accumulation of NP production capability in terms of BGC amount in certain taxonomic hot spots. The vast majority of Bacteroidetes BGC of the RiPPs, NRPS, PKS, and hybrid NRPS/PKS classes are unique compared with BGCs of any other phylum, in turn providing strong evidence of an overall high potential to discover novel scaffolds from these phylum's hot spots. Particularly, the genus *Chitinophaga* represents an outstanding group, on average encoding 15.7 BGCs per strain and enriched in NRPS and PKS BGCs. Based on this analysis, we selected 25 members of this genus for a cultivation and screening program. Our systematic chemotype-barcoding matrix pointed toward a tremendous chemical space of new NPs within this data set. Almost no known NPs were identified, strengthening the upfront-performed computational strain evaluation and selection. Eventually, this process led to the discovery of new

nonribosomally synthesized cyclic tetradeca- and pentadecalipodepsipeptides with iron chelating properties and candicidal activity. Those CLPs named chitinopeptins A, B, C1+C2, and D1+D2 were isolated from *C. eiseniae* DSM 22224 and *C. flava* KCTC 62435. The structures as well as the absolute stereochemistry of all amino acids were elucidated by extensive NMR studies and advanced Marfey's Analysis.

## RESULTS

**Bioinformatics analysis of the phylum Bacteroidetes.** The large and diverse phylum Bacteroidetes harbors Gram-stain-negative, chemo-organotrophic, non-spore forming rod shaped bacteria (47), graded into six so-called classes (48, 49). Members have colonized all types of habitats, including soil, ocean, freshwater, and the gastrointestinal tract of animals (38). Species from the mostly anaerobic *Bacteroidia* class are predominantly found in gastrointestinal tracts, while environmental Bacteroidetes belong primarily to the *Flavobacteriia*, *Cytophagia*, *Chitinophagia*, *Saprospiria*, and *Sphingobacteriia* classes (48, 49). Environmental studies based on amplicon diversity of adenylation and ketosynthase domains gave a first glance to the genetic potential of the phylum for the biosynthesis of NPs (46). In order to map the Bacteroidetes phylum systematically in terms of their BGC potential, we selected publicly available, closed, and annotated genomes in addition with some whole genome shotgun (WGS) projects at the time of data processing. In total, 600 genomes were analyzed using the "antibiotics and secondary metabolite analysis shell" (antiSMASH 5.0) (50). The determined total BGC amount as well as specific amount of NRPS, PKS, and hybrid BGCs was assigned to each single strain and set to the taxonomic context of the Bacteroidetes phylum, based on a phylogenetic tree calculated on complete 16S rRNA gene sequences (Fig. 1A). Assigning their BGC amount and types over the phylogenetic tree enabled comparisons between different classes and genera in terms of BGC amount and type. In most cases, a small linear positive correlation between genome size and number of secondary metabolite BGCs per genome is given, a phenomenon known from other bacteria (51) (Fig. 1A and B).

Mainly bacteria of the classes *Bacteroidia* and *Flavobacteriia*, with the smallest average genome size (3.78 and 3.51 Mbps) and an average BGC amount per strain (1.15 and 3.19 BGCs), display less significance for NP discovery. Exceptions are found in the genera *Kordia* (5.33 Mbps and 10 BGCs on average, three unique genomes analyzed [$n = 3$]) and *Chryseobacteria* (4.43 Mbps and 6.19 BGCs on average, $n = 48$) with up to five BGCs of the NRPS and/or PKS type. Many strains of these classes are pathogens and inhabit environments that are characterized by higher stability and lower complexity (e.g., guts) (38). NP production is an adaptive mechanism providing evolutionary fitness upon changing environmental conditions and in the presence of growth competitors (52). In accordance, the most talented bacterial NP producers, like the Actinomycetes (30) and Myxobacteria (53) are mainly found in highly competitive environments as e.g., soils. This correlation can also be seen within the Bacteriodetes phylum. In contrast to the anaerobic and pathogenic species, a higher BGC load was observed in the freely living and aerobic classes. The *Sphingobacteriia* and *Cytophagia* classes have an average genome size of 5.55 Mbps and 5.57 Mbps respectively, and an average BGC load of 5.79 and 4.63 per strain. An outlier is the genus *Pedobacter* with up to 22 BGCs on a single genome. Nevertheless, our analysis revealed that the *Chitinophagia* class outcompetes the other phyla members in respect to BGC amount per genome. Summarized, the class matched up with 11.4 BGCs per strain and genomes of an average size of 6.64 Mbps. Within this class, the genus *Chitinophaga* ($n = 47$) accumulates a enriched amount of 15.7 BGCs per strain on an average genome size of 7.51 Mbps. Thirty percent of their BGCs belong to the classes of NRPS and PKS, including rare *trans*-AT PKS BGCs. The genus with the second highest BGC load within the Bacteroidetes phylum is *Taibaiella* that also belongs to the *Chitinophagia* class, with a 27% smaller genome size and on average 19% less BGCs (Fig. 1C and D).

In order to discover novel chemistry, the pure amount of BGCs is of subordinate importance in comparison to the BGCs divergence, predicted to translate into structural

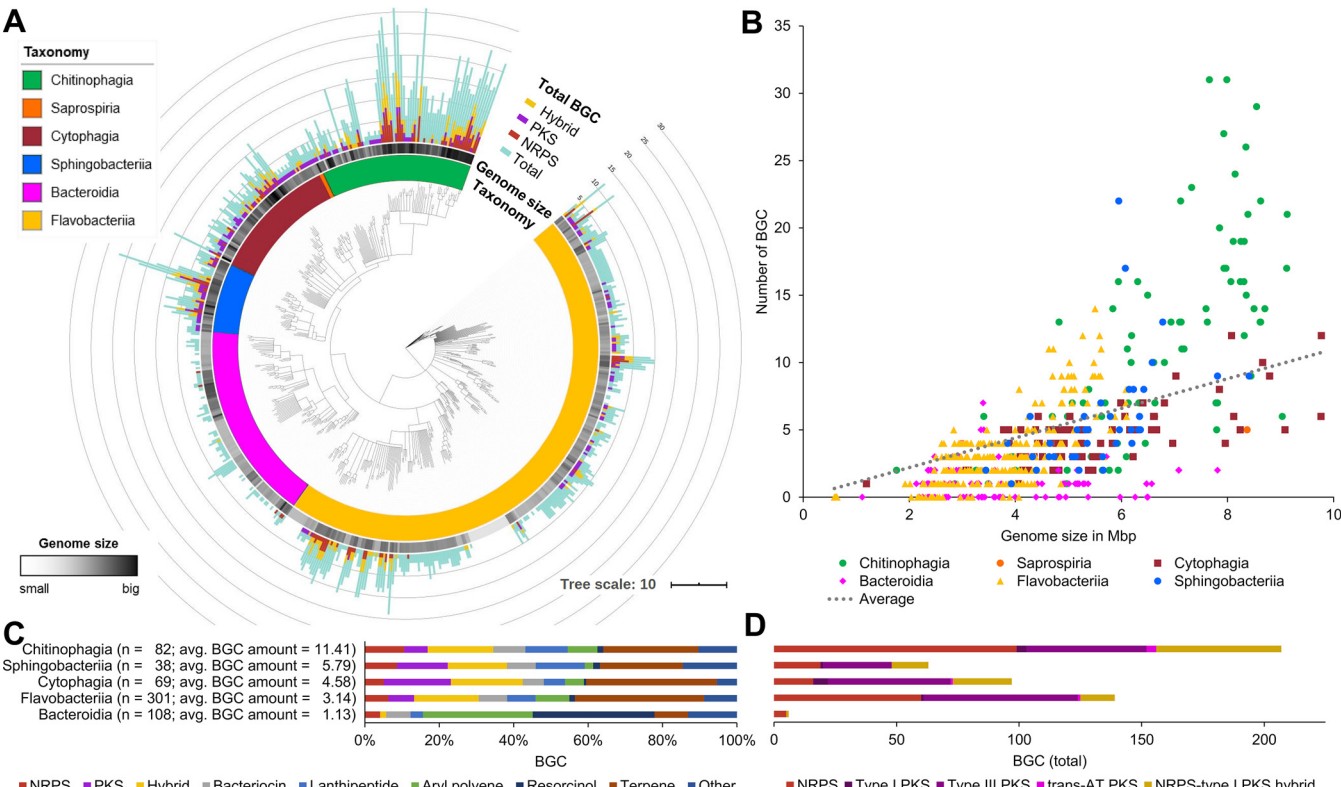

**FIG 1** Bioinformatics analysis of 600 genomes of the phylum Bacteroidetes points toward high biosynthetic genetic potential of the *Chitinophagia* class. (A) Consensus tree based on maximum-likelihood method (RAxML model v8 [108], GTR GAMMA with 1,000 bootstraps) of 600 16*S* rRNA gene sequences color coded on class level. For each strain the genome size and biosynthetic gene cluster amount and types are depicted. Tree is annotated using iTOL v4 (110). (B) Correlation of the total gene cluster amount with the genome size of each stain. (C) Analysis of the BGC types in the individual classes. BGC types: NRPS, nonribosomal peptide; PKS, polyketide; hybrid, cluster containing more than one BGC type; and other, remaining BGC types not separately listed. (D) Detailed look onto the most essential BGC types responsible for the production of bioactive NPs. Partial BGCs on contigs <10 kb of WGS projects are not included in all graphs.

diversity within the encoded metabolites (20, 54). Thus, we expanded the computational analysis by examining the sequential and compositional similarity of the BGCs detected in the 600 genomes using the "biosynthetic gene similarity clustering and prospecting engine" (BiG-SCAPE v1.0.0) (26). BiG-SCAPE creates a distance matrix by calculating the distance between every pair of BGC in the data set. The distance matrix combines three metrics, the percentage of shared domain types (Jaccard index), the similarity between aligned domain sequences (Domain sequence similarity) and the similarity of domain pair types (Adjacency index). The comparative analysis of the Bacteroidetes BGCs with the integrated MIBiG (Minimum Information about a Biosynthetic Gene cluster, v1.4) database (55) enabled their correlation to 1,796-deposited BGCs and consequently the correlation of their synthesized metabolites. In total, in 415 of the 600 genomes analyzed, 2,594 BGCs were detected and grouped with a default similarity score cutoff of c = 0.6 into a sequence similarity network with 306 gene cluster families (GCFs). Only 12 GCFs clustered with MiBIG reference BGCs of known function. Together, those 12 GCFs comprise 11.5% (298 BGCs) of all detected Bacteroidetes BGCs. Nine of them belonged to the BiG-SCAPE BGC classes of "PKSother," "Terpenes," or "Other." These GCFs encode known NP classes like biotin, ectoine, *N*-acyl glycin, and eicosapentaenoic acid, as well as products of the NRPS-independent siderophore (NIS) synthetase type, precisely desferrioxamine and bisucaberin B (56), forming two connected though distinct clouds (Fig. 2). A Cytophagales specific GCF of the terpene class includes the MIBiG BGC0000650, encoding the carotenoid flexixanthin (57). The Bacteroidetes are well known producer of flexirubin-like pigments (aryl polyenes), which is reflected in a conserved biosynthesis across several genera (58, 59). The flexirubin gene cluster cloud (GCC) covers 268 BGCs from 252

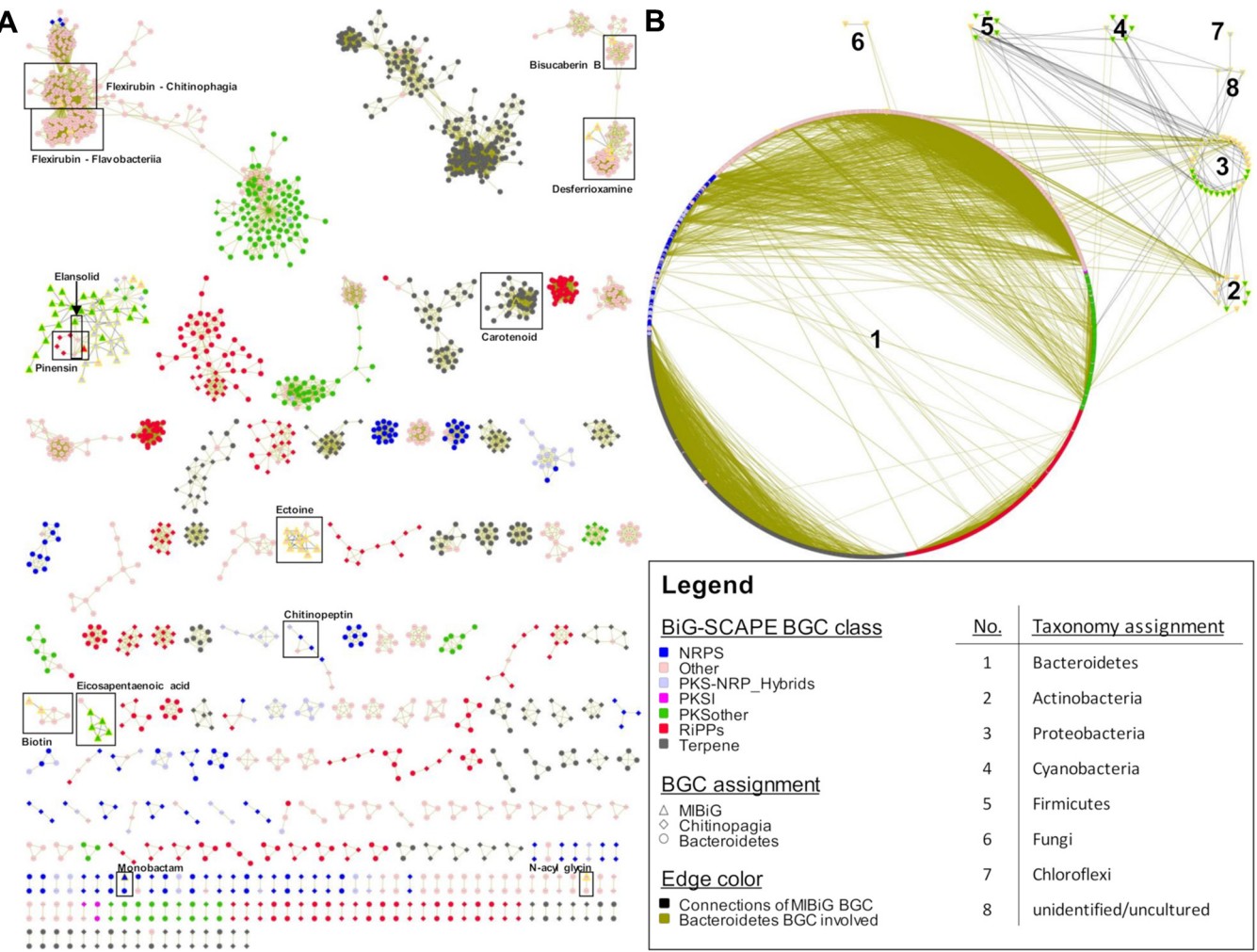

**FIG 2** BiG-SCAPE (26) analysis of the phylum Bacteroidetes highlights a giant uncovered genetic potential. (A) A global network of all depicted gene cluster families with a cutoff of 0.6. Known ones are marked, named and are identified by known biosynthetic gene cluster (BGC) deposited at MIBiG (all known and deposited Bacteroidetes BGCs are found). (B) Same BGCs sorted by taxonomy. BiG-SCAPE BGC classes: NRPS, nonribosomal peptide; PKS, polyketide; and RiPPs, ribosomally synthesized and posttranslational modified peptides. Singeltons (unique BGCs without any connection) are not shown. Visualization and manipulation by Cytoscape v3.4.0 (113).

individual strains and five of six analyzed classes. In our analysis, only the newly formed *Saprospiria* class (49) was an exception. However, considering that the analysis included only two *Saprospiria* strains it does not yet allow any integral assessment of its capabilities to produce these yellow pigments. The flexirubin GCC can be divided into at least five distinct GCFs. Resolved on class level this revealed a specific *Flavobacteriia* family including BGC0000838 from *Flavobacterium johnsoniae* UW101 (58) as well as a specific *Chitinophagia* family including BGC0000839 from *Chitinophaga pinensis* DSM 2588 (59). The latter in turn being directly connected to a third GCF, in majority covering BGCs from its genus *Chitinophaga* while not including a reference BGC.

In addition, the reference BGCs described to encode the bioactive NPs monobactam SQ 28,332 (60, 61), elansolids (40, 62), and pinensins (41), are annotated to three distinct GCFs (Fig. 2A). The monobactam BGC (BGC0001672) was unique and only identified in its described producer strain *Flexibacter* sp. ATCC 35103. Elansolids and pinensins represent patent protected chemical entities active against Gram-positive bacteria and filamentous fungi and yeasts, respectively. The complete elansolid encoding BGC (BGC0000178), almost 80 kbp in size, was identified in the genomes of strain *Chitinophaga* sp. YR627 and *Chitinophaga pinensis* DSM 2588 (=DSM 28390 [63]) with the genetic potential to produce elansolid already proposed for *C. pinensis* (64) (Fig. S1A). Besides the original producer strain, *Chitinophaga sancti* DSM 21134 (65), these

strains provide alternative bioresources to access these polyketide-derived macrolides. In addition, both strains also harbor the pinensin BGC directly co-localized with the elansolid-type BGC. This co-localization leads to an artificial connectivity between both GCFs by using the chosen BiG-SCAPE parameters. Manual curation revealed six strains carrying a pinensin-like BGC in their genome in total (Fig. S1B). The alignment of the RiPP core peptide revealed that only the amino acid sequence from strain *Chitinophaga* sp. YR627 was identical to the described pinensin sequence. The other strains show amino acid sequence variations, pointing toward structural variance (Fig. S1C).

With >200 GCFs identified and only 12 of them annotated toward known BGCs and their metabolites, the sequential and compositional similarity analysis revealed a BGC diversity within the Bacteroidetes phylum, differing from the composition of known BGCs deposited in the MiBIG database.

Extension of this similarity network analysis toward taxonomic relations on phylum level showed that Bacteroidetes BGC of the RiPPs, NRPS, PKS, and hybrid NRPS/PKS classes are unique compared with BGCs of any other phylum (Fig. 2B). This in turn provides a strong evidence of a general high potential to discover novel metabolites from this phylum. The majority (~66%) of all GCFs belonging to the above-mentioned BGC classes are found in the *Chitinophagia* class, only representing 13.7% of the analyzed strains. Within this class, the genus *Chitinophaga* can be prioritized in terms of BGC amount and composition. In respect to many more complete unique and novel RiPP, NRPS, PKS, and hybrid BGCs thereof (not depicted in the network), this is a strong indication that the biosynthetic potential within this genus is far from being fully exploited. It can be considered as the most promising starting point for the discovery of novel metabolites within this phylum.

**Metabolomics of the *Chitinophaga*.** Based on the genomic data evaluation, we selected the *Chitinophaga* for performing a bioactivity guided NP discovery program. NPs are considered to be nonessential metabolites for bacterial growth and reproduction but rather providing evolutionary fitness, thus, being expressed as adaptive response to changing environmental conditions. Consequently, the discovered BGC potential is not expected to translate into the actually produced metabolite pattern under laboratory conditions (22). A common theme of strategies to approach this challenging link is the cultivation in several media variants exposing the strains to various stress conditions, e.g., nutrient depletion (66, 67). As a consequence of nutrient depletion, bacteria enter the stationary phase and reduce or even cease growth, often found to coincide with induction of secondary metabolite production (66, 68). To trigger these events, we cultivated a diversity of 25 *Chitinophaga* strains (Table S1) in five different media for 4 as well as 7 days.

The metabolites were extracted from freeze-dried culture broths with methanol and the organic extracts were subsequently analyzed by ultra-high performance liquid chromatography-high resolution mass spectrometry (UHPLC-QTOF-HR-MS). LC-MS data sets from a total of 250 extracts (and media controls) were examined allowing the definition of strain-specific molecular features. In an initial step, features (represented by *m/z*, retention time, isotope pattern) were calculated within all extracts. Curation of all data sets was necessary to filter background noise and confirm the authenticity of defined features. This curation step helped to avoid false uniqueness due to concentrations near the corresponding detection limit and to reduce the possibility of picking up background noise. Furthermore, the possibility of multiple mass spectrometric features for any NP contributes to the complexity of those data sets, e.g., by the formation of different ion adducts and in-source-generated fragment ions of single molecules. The final data set consisted of 93,526 features. Those were aligned into 4,188 buckets with a bucket being defined as an *m/z* and retention time (RT) region hosting all features with matching *m/z* and RT (69). We created a chemotype-barcoding matrix of this complex data set, allowing its visualization and evaluation (Fig. 3A). After normalization of the data set by buckets congruent with the media controls (in total 1,452),

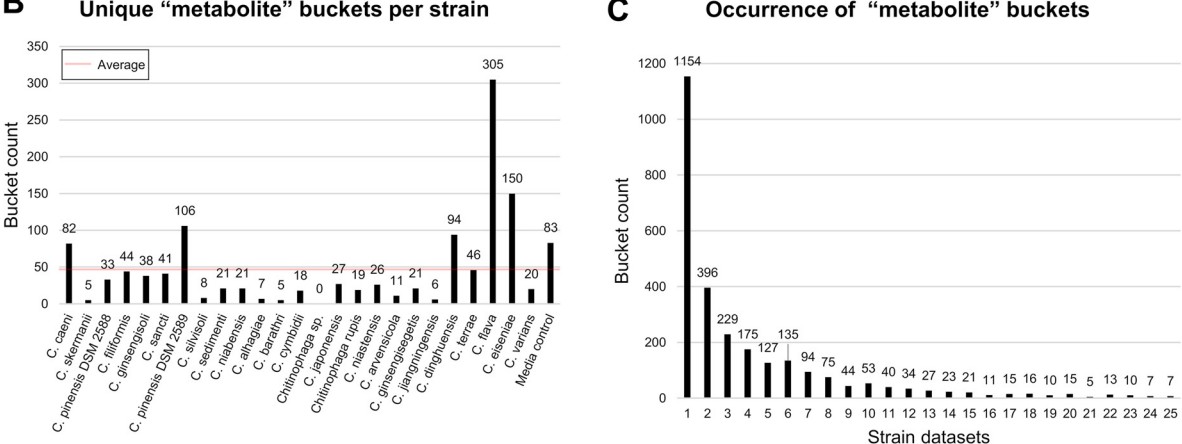

**FIG 3** Taxonomical arrayed chemotype-barcoding matrix reveals an uncharted chemical space within the genus *Chitinophaga*. (A) The tree is based on a Clustal W alignment (111) of available 16S rRNA gene sequences of 25 *Chitinophaga* strains available for cultivation. The tree was calculated using MEGA v7.0.26 with the maximum-likelihood method and GTR-Gamma model (112). Percentage on the tree branches indicate values of 1,000 bootstrap replicates with a bootstrap support of more than 50%. The tree is drawn to scale, with branch lengths measured in the number of substitutions per site. The bucketing process is depicted as a chemotype-barcode matrix and color coded by the condition each individual bucket was present. (B) Bar plot of the unique "metabolite" buckets of each strain. (C) Bar plot depicting the occurrence of "metabolite" buckets in the data sets.

we determined in total 2,736 buckets as representing produced metabolites of the investigated *Chitinophaga* set.

The detected buckets were analyzed for presence in all combinations considering utilized cultivation media and incubation time. They were put in order according to 16S rRNA sequence similarity on strain level. In order to facilitate data interpretation

and to identify strain specific as well as conserved metabolites, all buckets were sorted according to their frequency of appearance. Comparative visualization of the short (4 days) and prolonged (7 days) incubation time revealed differences on the chemotype profile of the strains. Most strains metabolize the majority of media components within the first 4 days of cultivation, recognized by only a small number of "media buckets" still present in the extracts after this incubation period. A second population of rather slow-growing strains appeared to shift the metabolic profile only after 7 days of incubation in comparison to the respective media control (e.g., *C. caeni* KCTC 62265, *C. dinghuensis* DSM 29821, *C. niastensis* DSM 24859, *C. barathri*, and *C. cymbidii*). Especially *Chitinophaga* sp. DSM 18078 required a prolonged incubation time to metabolize the media ingredients, producing 77.4% more metabolite buckets after 7 days of incubation in comparison to the earlier sampling time. Within the whole data set, on average 26.6% (92.4) more metabolite buckets were detected after seven than after 4 days of incubation. In contrast, a fraction of bacterial metabolite buckets disappeared within the extracts of five strains after prolonged incubation, showing the necessity to vary cultivation conditions to access a possibly comprehensive metabolite profile of each investigated strain. The many media-specific buckets (colored) compared with the ones being produced in various media (black) also show the effect of variations of the bacterial nutrient supply (Fig. 3A). In combination with the varied cultivation period, this led to an average of approximately 46 unique buckets per strain. Especially *C. flava* KCTC 62435 and its close relative *C. eiseniae* DSM 22224 outcompeted the others by producing 305 and 150 unique metabolite buckets respectively (Fig. 3B), indicating their status as best-in-class producers. In total 1,154 buckets (~42%) of the entire data set were identified only in one respective strain data set, representing strain-specific metabolites (Fig. 3C). This high level of strain-specific metabolites shows the heterogeneity of the investigated strain set. In parallel, this experiment depicts a structural diversity in terms of molecular size up to 5145.051 Da and covered polarity range (Fig. S2).

Analysis of the conserved metabolites (which includes primary and secondary metabolites) revealed 357 buckets to be present in at least 10 out of the 25 total analyzed strains while only the low number of seven buckets could be detected within samples of all *Chitinophaga* strains investigated. Based on their MS$^2$-fragmentation patterns sharing the loss of long carbon chains, we postulated a structural relationship between four of them and assigned them as hitherto unknown (amino/phospho) lipids.

Next, the complete LC-MS data set was examined for the presence of structurally characterized microbial NPs (~1,700) deposited in our in-house database on the basis of accurate *m/z*, RT, and isotope pattern. The frequency of rediscovery was zero. Considering the known bias of the database for natural products from classical NP producing taxa such as Actinobacteria, Myxobacteria, and fungi, this confirmed the low congruency toward these taxa, which was also found by our BGC categorization study. A complementary scan of LC-MS/MS data of the entire data set was compared with *in silico* fragments of >40k NPs deposited in the commercial database AntiBase (70). Congruently, no database-recorded NP was identified within our data set besides falcitidin (71), an acyltetrapeptide produced by a *Chitinophaga* strain. Although this finding shows the general applicability of this workflow, the underrepresentation of NPs isolated from the phylum Bacteriodetes also in public databases is still a severe limitation for comprehensive categorization of their omics-data today.

However, as a consequence, these data provided a high confidence level in the investigated strain portfolio and showed that the metabolite spectrum produced by these strains is largely underexplored and different from the metabolites produced by classical NP producer taxa. Even though strains of the genus *Chitinophaga* are phylogenetically closely related, they produce a heterogeneity of metabolites; thereby, showing a high number of strain-specific metabolites, associated with a likelihood for chemical novelty.

**Chitinopeptins, new CLPs from *C. eiseniae* and *C. flava*.** A correlation of those untapped buckets with antibacterial activity was performed by screening the organic crude extracts against a panel of opportunistic microbial pathogens. In particular, methanol extracts of *Chitinophaga eiseniae* DSM 22224 and *Chitinophaga flava* KCTC

**FIG 4** Chemical structures of the chitinopeptins A–D. Compounds 1 and 2 are produced by *C. eiseniae* DSM 22224, compounds 3 to 6 by *C. flava* KCTC 62435.

62435, the two strains with the highest level of metabolic uniqueness, exhibited strong activity against *Candida albicans* FH2173. Bioactivity assay and UHPLC-HR-ESI-MS guided fractionation led to the identification of six new cyclic lipodepsipeptides. *C. eiseniae* produced the two tetradecalipodepsipeptides chitinopeptins A and B (1 and 2) with molecular formulae $C_{82}H_{137}N_{17}O_{28}$ (1, $[M+H]^+$ 1809.0052) and $C_{81}H_{135}N_{17}O_{28}$ (2, $[M+H]^+$ 1794.9907). Whereas *C. flava* assembled the four pentadecalipodepsipeptides chitinopeptins C1+C2 and D1+D2 (3 to 6) with molecular formulae $C_{84}H_{140}N_{18}O_{30}$ (3 and 4, $[M+H]^+$ 1882.0177) and $C_{83}H_{138}N_{18}O_{30}$ (5 and 6, $[M+H]^+$ 1868.0059) (Fig. 4). All

compounds were present in the MS spectra as pairs of $[M + 3H]^{3+}$ and $[M + 2H]^{2+}$ ions (Fig. S3A). All six native peptides appeared to be highly stable, because only poor yields of fragment ions arose using electrospray ionization source (Fig. S3B), even under elevated collision energy conditions (up to 55 eV), thereby preventing MS/MS based structure prediction and structural relationship analysis using molecular networking.

The chemical structures of the six compounds were determined by extensive NMR studies using 1D-$^1$H, 1D-$^{13}$C, DQF-COSY, TOCSY, ROESY, multiplicity edited-HSQC, and HMBC spectra (Fig. 4). The analysis of the compounds in most "standard" NMR solvents was hampered by either extreme line broadening (DMSO, MeOH, pyridine) or poor solubility ($H_2O$, acetone). However, a mixture of $H_2O$ and $CD_3CN$ (ratio 1:1) gave rise to NMR spectra of high quality and confirmed the presence of peptides. In order to obtain a good dispersion of the amide resonances, $^1$H-spectra were acquired at different temperatures between 290 and 305 K. A temperature of 299 K or 300 K, respectively, was found to be the best compromise considering signal dispersion and line broadening (Fig. S4 to 31 and Table S2 and S3).

The first compound to be studied was chitinopeptin A. The analysis of the NMR spectra revealed the presence of several canonical amino acids (1 Thr, 1 Ala, 1 Ile, 1 Ser, 1 Lys, and 2 Leu), and several hydroxylated amino acids (3 $\beta$-OH Asp, 1 $\beta$-OH Phe, 1 $\beta$-OH Ile). In addition, one $N$-methyl Val and one 2,3-diaminopropionic acid (Dap) moiety could be assigned. The sequence of the amino acids was established by correlations in the ROESY ($NH_i/NH_{i+1}$, $NH_{i+1}/H\alpha_i$) and HMBC spectrum ($C'_i/NH_{i+1}$). The formation of a cyclic peptide was indicated by the $^1$H-chemical shift of the $\beta$-proton of the Thr residue in position 2 (5.23 ppm) and the correlation in the HMBC spectrum between the carbonyl carbon of the $C$-terminal $\beta$-OH Ile (position 14) and the $\beta$-proton of the Thr residue.

Aside from the amino acids, a modified fatty acid residue was identified which could be described as 2,9-dimethyl-3-amino decanoic acid. Correlations in the HMBC spectrum between the carboxyl carbon (C1) and the $N$-methyl group of the $N$-methyl Val proved its position at the $N$-terminus of the peptide. The structure of chitinopeptin B was almost identical to the structure of CLP 1. The only difference was the substitution of the 2,9-dimethyl-3-amino decanoic acid by 3-amino-9-methyl decanoic acid.

CLPs 3 and 4 were isolated as a 5:4 mixture of two components. One of the main differences compared with the structures above was an additional Dap residue, which was inserted between the fatty acid moiety and $N$-methyl Val. Both components contain an Ile instead of a Leu (CLPs 1 and 2) at position 10. Furthermore, the 2,9-dimethyl-3-amino decanoic acid is replaced by a 3-hydroxy-9-methyl decanoic acid. The two components 3 and 4 differ in the constitution of the Dap in position 9. In one component 4, the peptide bond between the $\alpha$-amino function and the carbonyl group of Lys is formed, while in the other component 3, the $\beta$-amino group (side chain) is connected to the carbonyl group of Lys. The same pair of structures as for CLPs 3 and 4 is obtained in the case of CLPs 5 and 6. In contrast to the previous ones, both components contain a Val in position 10 instead of Leu or Ile, respectively.

The absolute stereochemistry of the amino acids in the CLPs was determined by using advanced Marfey's Analysis (72). Comparison of the RTs with (commercially available) reference amino acids allowed identification of nine out of 14 (CLPs 1 and 2) and 10 out of 15 amino acids (CLPs 3–6), respectively. RTs of $N$-methyl-L-Val, D-allo-Thr, D-Ala, D-allo-Ile (2 × for CLPs 3 and 4), L-Ser, D-Lys, L-Dap (2x for CLPs 3 to 6), D-Leu (only CLPs 1 and 2), L-Leu, and D-Val (only CLPs 5 and 6) matched the reference ones. Assigning L- as well as D-leucine within structures 1 and 2 to position 13 and 10, respectively, was possible because position 10 was the only variable position in all six depsipeptides. Either Ile (CLPs 3 and 4) or Val (CLPs 5 and 6) were identified at this position with all amino acids having D-configuration. Therefore, it can be assumed that D-Leu is present at position 10 in CLPs 1 and 2 (Fig. S32 and 33).

Authentic samples of the $\beta$-hydroxyamino acids or suitable precursors were synthesized utilizing modified literature known procedures. All four stereoisomers of $\beta$-hydroxyaspartic acid were obtained from (−)-dibenzyl D-tartrate or (+)-dibenzyl L-tartrate,

respectively, according to a procedure described by Breuning et al. (73). While the *anti*-isomers are directly accessible, the *syn*-isomers were obtained by selective base-induced epimerization of the azido-intermediates and separation of the two isomers by HPLC. Cbz-protected L-isomers of the $\beta$-hydroxyphenylalanines and $\beta$-hydroxyisoleucines were synthesized starting from an orthoester protected L-serine aldehyde, initially described by Blaskovich and Lajoie (74–76). To cover the corresponding D-isomers for analytical purposes, racemic samples of the amino acids were produced by racemization of the orthoester protected L-serine aldehyde by simple chromatography on silica (74). Advanced Marfey's Analysis determined (2*S*,3*S*)-3-hydroxyaspartic acid, (2*S*,3*R*)-3-hydroxyphenylalanine and (2*S*,3*R*)-3-hydroxyisoleucine as the absolute stereochemistry of $\beta$-hydroxyamino acids for all six CLPs (Fig. S34 to 36).

To the best of our knowledge, these represent the first CLPs described from the genus *Chitinophaga* and after the recently described isopedopeptins (12), the second CLP family of the entire phylum. They contain a high number of non-proteinogenic amino acids (i.e., 12 of 14 amino acids in CLPs 1 and 2, and 13 of 15 in CLPs 3 to 6). Beta-hydroxylations of Asp, Phe, and Ile are the most abundant modifications and *N*-methyl Val and Dap are incorporated into the peptide backbone. Furthermore, during LC-MS analysis, a mass shift of 52.908 Da accompanied with an emerging UV maximum at 310 nm, as well as a shift in RT was observed for compounds 1 to 6. This traced back to the coordination of the compounds to iron impurities during LC-analysis, which was confirmed by the addition of Fe(III)-citrate to the compounds prior to LC-MS analysis (Fig. S37). We postulated that the RT shift is due to a conformational change and altered polarities as a consequence of iron complexation. Iron coordination is a known feature of siderophores, produced by bacteria upon low iron stress (77). To investigate the impact of iron on the CLPs production, CLP 1*C. eiseniae* was cultured in 3018 medium (for composition see Materials and Methods section) supplemented with different iron concentrations. However, the overall production of CLPs 1 and 2 and their iron complexes was not repressed by increased iron levels, contrasting the iron-responsive productivity of classical siderophores (Fig. S38).

Chitinopeptins A to D were tested against eight Gram-negative and five Gram-positive bacteria, as well as against three filamentous fungi and *C. albicans* (Table S4). For all tested CLPs, activity was observed against *M. catarrhalis* ATCC 25238 and *B. subtilis* DSM 10, exhibiting MICs down to 2 $\mu$g/mL. The tetradecalipodepsipeptides 1 and 2 exhibited activity at 4 to 8 $\mu$g/mL against *C. albicans* FH2173, while the pentadecalipodepsipeptides exhibited MICs of only 16 $\mu$g/mL. Screenings against filamentous fungi revealed MICs of 16 $\mu$g/mL against *Z. tritici* MUCL45407, while no activity was observed against *A. flavus* ATCC 9170 and *F. oxysporum* ATCC 7601. To investigate the impact of iron-binding on the bioactivity, CLP 1 was tested as a representative also in its iron complexed form confirmed by LC-MS analysis (Fig. S37). This revealed that the bioactive potency of the iron complex is reduced in comparison to its iron free form, although not completely suppressed (Table S4).

**BGCs corresponding to chitinopeptins.** In order to identify the BGCs encoding the chitinopeptins' biosynthesis, we scanned the genomes of *C. eiseniae* (FUWZ01.1) and *C. flava* (QFFJ01.1) for NRPS-type BGCs matching the structural features of the molecules. The number and predicted substrate specificity of the A-domains, the overall composition of the NRPS assembly line, as well as precursor supply and post-assembly modifications were taken into account. We identified the BGCs in *C. eiseniae* and *C. flava* congruent to the CLP structure in each case (Fig. 5A). Furthermore, the positioning of all epimerization domains within the detected NRPS genes, encoding the conversion of L- to D-amino acids, is in agreement with the determined stereochemistry of the molecules. Thereby, the domains are classically embedded in the NRPS assembly lines. No racemase(s) encoded in *trans* or C domains catalyzing the conversion of amino acids, are observed as it is the case e.g., in the BGC of the stechlisins, CLPs produced by *Pseudomonas* sp. (78).

In our BiG-SCAPE network, these BGCs from *C. eiseniae* and *C. flava* were part of a GCF, and manual inspection confirmed two further related BGCs within the genomes of *C. oryzi-terrae* JCM16595 (WRXO01.1) and *C. niastensis* DSM 24859 (PYAW01.1) (Fig. 5B). Besides the

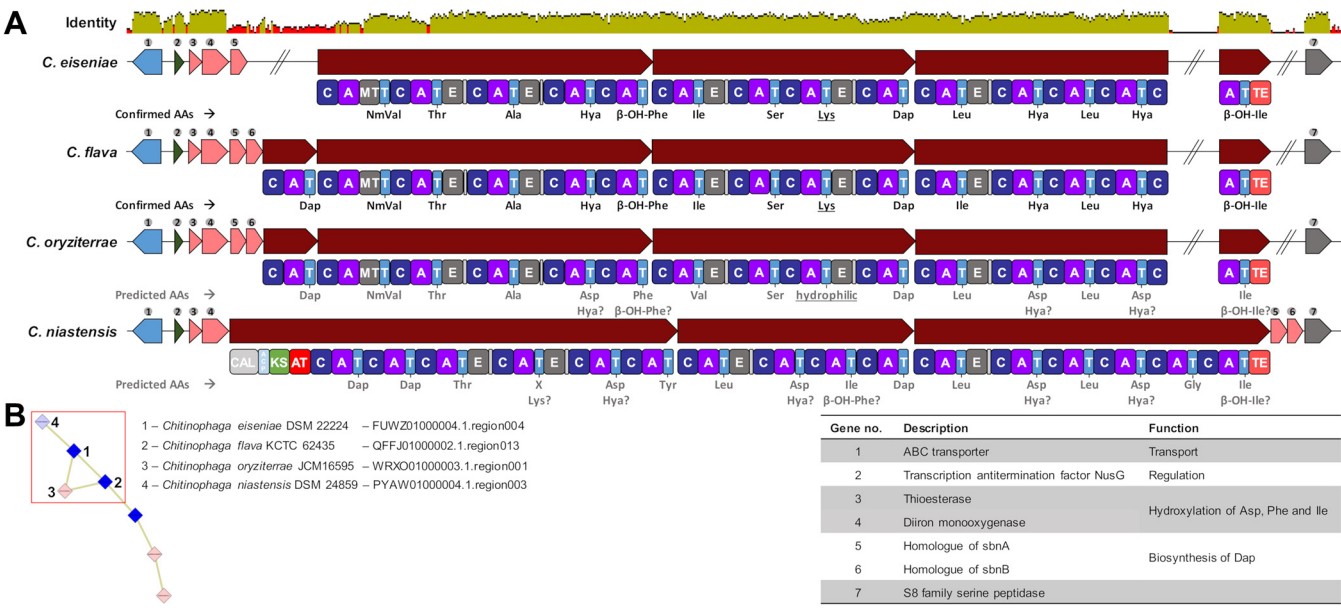

**FIG 5** Biosynthetic gene clusters responsible for the production of chitinopeptin A to Dand further derivatives. (A) BGCs of strains *C. eiseniae*, *C. flava*, *C. oryziterrae*, and *C. niastensis* with matching amino acid sequence and additional biosynthetic genes responsible for the hydroxylation of Asp, Phe, and Ile or the in cooperation of 2,3-diaminopropionic acid. Identity was calculated with a standard MAFFT alignment (107). (B) Gene cluster family of this iron chelating cyclic lipodepsipeptides. AA, amino acids; NmVal, *N*-Me-Val; Hya, *β*-hydroxyaspartic acid; C, condensation domain; A, adenylation domain; MT, nitrogen methyltransferase; T, peptidyl-carrier protein domain; E, epimerization domain; TE, thioesterase domain; CAL, co-enzyme A ligase domain; ACP, acyl-carrier protein domain; KS, ketosynthase domain; AT, acyltransferase domain.

structural NRPS genes, further genes are conserved between all four BGCs, predicted to encode an ATP-binding cassette (ABC) transporter, a transcription factor, an S8 family serine peptidase, a thioester reductase domain, and a metal *β*-lactamase fold metallo-hydrolase. The biosynthesis of these CLPs requires a *β*-hydroxylation tailoring reaction of the precursor Asp, Phe, and Ile. This structural feature is also present in chloramphenicol and its biosynthesis was shown to be catalyzed by the diiron-monooxygenase CmlA that catalyzes substrate hydroxylation by dioxygen activation. CmlA coordinates two metal ions within a His-X-His-X-Asp-His motif and possesses a thioester reductase domain (79). Both features are also present in all four detected BGCs, encoded by two separate genes annotated as thioester reductase domain and metal *β*-lactamase fold metallo-hydrolase (gene 3 and 4) (Fig. 5B and Table S5). Moreover, all four BGCs contain genes with sequence similarity to *sbnA* and *sbnB*, encoding enzymes that catalyze the synthesis of the non-proteinogenic amino acid Dap (80, 81) (Table S6). The presence of genes encoding for supply with precursor amino acid(s) is a common feature in BGCs corresponding to NPs described in the Firmicutes and Actinobacteria phyla (82–87). Interestingly, within the BGC of *C. eiseniae* a *sbnB* homologue is missing. To rule out an error during genome sequencing, assembly, and annotation, we amplified the respective section by PCR and confirmed the published genome sequence. However, not encoded in *cis*, *C. eiseniae* carries further *sbnA* and *sbnB* homologues. These are encoded in other NRPS-type BGCs (FUWZ01000006, location: 555,550 to 557,479 and FUWZ01000004, location: 273,583 to 275,522) and potentially function in *trans* to compensate the absence of the gene within the chitinopeptin A and B BGC. In general, we observed that the Dap subcluster *sbnA/B* is an abundant genetic feature of Bacteroidetes secondary metabolism, present in many BGCs. This is not restricted to *Chitinophaga*, but expanded to different genera and in consequence, a structural feature of Bacteroidetes NPs predictively found with high frequency. Indeed, Dap moieties are present in the known antibacterial Bacteroidetes compounds isopedopeptins (12) and TAN-1057 A to D (43).

Considering the composition of the structural NRPS genes in terms of A domain number as well as predicted A domain substrate specificity (88), we propose that *C. oryziterrae* and *C. niastensis* carry the potential to provide additional structural variety

to the chitinopeptins (Table S7). The NRPS gene of *C. niastensis* encodes 16 A-domain-containing modules, predictably producing hexadecapeptides. Furthermore, an initial PKSI module replaces the C-starter domains present in the three other BGCs of this GCF. This points toward to an attachment of carboxylic acid residues to the peptide scaffold, resulting in further compound diversification. While *C. oryziterrae* was not available for cultivation, we inspected the extracts of *C. niastensis* for putative further CLPs. Indeed, although only detected in traces, possible products could be identified based on comparable RTs and isotope patterns with *m/z* of 680.9817 [M + 3H]$^{3+}$ and 685.6545 [M + 3H]$^{3+}$ (Fig. S39).

## DISCUSSION

Mining microbes for their secondary/specialized metabolites continues to be an essential and valuable part of drug discovery pipelines. With the constant rediscovery of already known NPs, the past decades are representative for the limitations of classical approaches focusing on the most talented producer taxa that only represent a limited phylogenetic space of the bacterial kingdom (34). However, the microbial diversity is growing with each new phylum added to the tree of life (20). In parallel, rapid improvements of resolution and accuracy allowed us to apply technologically more complex metabolomics- and genomics-guided methods to mine nature (24, 25, 31, 33). Genomics-guided methods predict the strains phenotype and their theoretical ability to produce metabolites, based on identified BGCs and their consecutive prioritization (23). Complementary omic-technologies, especially metabolomics enable the sophisticated characterization of the strains' chemotypes, the actually detected metabolites. Standalone or applied sequentially, both technologies already made a strong impact on the field of NPs. (Semi)automatized combinations of genomics and metabolomics are under development and aim to establish a direct link between the biosynthetic potential of microbial strains and their actually expressed metabolites (29, 33, 89). The basis for this was built over the last years and gave evidence that those tools are as powerful as their underlying training data sets (90).

In the future, these approaches will likely support the field by promoting reduction of rediscovery rates and in turn allowing efficient sample prioritization with respect to the target molecules of the respective studies (91). Until today, these approaches were primarily applied on big data sets from well-known NP producing taxa such as Actinobacteria or Myxobaceria (29, 30). These studies enabled technology development, while directly expanding the overall knowledge on the still available chemical space of these taxa as well as the identification of novel secondary metabolites or derivatives of known ones. Here, we translated this approach of application of complementary omics-technologies for the first time to the currently still underexplored Bacteroidetes phylum. The comprehensive computational analysis of publicly available Bacteroidetes genomes revealed the *Chitinophaga* as the most promising genus in terms of NP production capabilities. The genome-wide evaluation of 47 *Chitinophaga* strains showed the presence of 15.7 BGCs on average, with a minimum of six and a maximum of 31 BGC. Although the number of detected BGCs is strongly dependent on the utilized bioinformatics tools and chosen settings, the most talented *Chitinophaga* competed with the BGC loads of some Actinobacteria (92). Moreover, the variation of BGC amount within a genus is a known phenomenon and also observed e.g., in the *Amycolatopsis* genus (93).

The Dictionary of Natural Products (DNP) (94) (status 01/2021) contains only 14 entries ascribed to the genus *Chitinophaga*. This does not reflect their predicted genetic potential, because e.g., for the genus *Streptomyces*, 8,969 chemical entities are deposited. The extrapolation of the approximately doubled BGC load of *Streptomyces* (25-70 BGCs) (51) results in ~650 times more chemical entries, a theoretical gap that justifies efforts to investigate the metabolic repertoire of the genus *Chitinophaga*. Our metabolomics analysis of 25 *Chitinophaga* strains confirmed the genomic and database-based prediction by empirical data. The identified chemical space is represented by more than 1,000 unique and unknown candidates that could not been associated to any microbial NP known today. Specialized metabolites do not

have a pivotal role in the survival of the microorganism. They are considered not essential for vegetative growth, as well as reproduction, but to give a fitness advantage in specialized growth conditions. As a consequence of that, these metabolites are not constantly expressed, but rather as a specific response to defined environmental conditions. Their production occurs generally in a growth-phase dependent manner and often coinciding with reduction in bacterial growth rate or even growth cessation mainly based on nutrient depletion (68). The strategy to challenge the microorganisms by varying cultivation conditions and incubation times successfully translated also in the genus *Chitinophaga* toward an increased metabolite diversity. This shows the need to adjust the cultivation time to the specific growth pace of the investigated strains. The uniqueness and value of this data set compares with data sets gathered by metabolomics studies of well-known NP producer taxa. Although comparability of such extensive approaches is challenging due to different data set size, composition as well as technical devices and settings, the detection of bucket/metabolite numbers per strain in our *Chitinophaga* strain set shows a similar range compared with the ones described for different Myxobacteria genera (32). One striking difference however, is the lack of dereplicated known metabolites, which can be explained by their underrepresentation in all reference databases utilized in this study for data categorization. This is likely due to the fact that investigations of the genus *Chitinophaga* and the Bacteroidetes phylum itself have just started.

In conclusion, our analysis reveals that the Bacteroidetes phylum has a distinct metabolic repertoire compared with classical NP producer taxa, indicated by the small overlapping taxonomic relationship of their GCFs. Eventually, our combinatorial approach in utilizing genomics and metabolomics facilitated the genus and strain prioritization and paved the way for the discovery of the chitinopeptins A to D. These NRPS-assembled CLPs exhibit primarily activity against *Candida albicans* and were found to coordinate iron. Binding of metal ions is a feature also described for other CLPs such as pseudofactin II, which displays an increased antimicrobial activity upon metal-coordination due to disruption of the cytoplasmic membrane in its chelated state (95). In general, it is known that individual CLPs possess multiple functions such as iron chelation, antimicrobial activity, and interaction in bacterial motility (19).

Incidences of fungal infections, including candidiasis caused by *Candida* species, are increasing (96). Currently, clinically used antifungals are limited to four classes: azoles, polyenes, echinocandins, and pyrimidine analogs. Associated with occurrence and spreading of resistances, this leads to alarmingly decreased treatment success and increased possibilities of fatal outcomes (97). This development clearly demands counteraction. One possible way to use is the constant discovery of NPs with novel antifungal activity. The chitinopeptins share properties with the NP family of CLPs that exhibit an intrinsic antifungal activity. Mode of action studies for these compounds suggest a membrane interaction and leakage effects up to pore-forming properties (98–101). Although certain pharmacological properties, such as metabolic stability and cell permeability, represent major challenges for drug development (102), their application as e.g., biocontrol agents and food preservatives is under investigation or already reached commercialization (16, 103). The chitinopeptins thereby exhibit similar activities against *Candida* as CLPs of the surfactin or iturin (104) and fengycin (15) group. Further studies are necessary to investigate their specific activity profile.

## MATERIALS AND METHODS

**Genomic data processing.** Genomic data were collected until January 2021. To that date, almost all publicly, complete, and annotated genomes in addition with some WGS projects (<100 scaffolds) of the phylum Bacteroidetes (600 genomes in total, Table S8) were analyzed using antiSMASH 5.0 (50) and BiG-SCAPE v1.0.0 (26). All Bacteroidetes genome sequences used in this study were downloaded from the National Center for Biotechnology Information (NCBI) database or the Department of Energy (DOE) Joint Genome Institute–Integrated Microbial Genomes & Microbiomes (JGI IMG) database (105, 106). The data set consists of all six phylogenetic classes with the following distribution: *Bacteroidia* 18% (*n* = 108), *Chitinophagia* 13.7% (*n* = 82), *Cytophagia* 11.5% (*n* = 69), *Flavobacteriia* 50.2% (*n* = 301), *Saprospiria* 0.3% (*n* = 2), *Sphingobacteriia* 6.3% (*n* = 37).

**Phylogenetic tree reconstruction.** For the phylogenetic tree of the complete Bacteroidetes phylum, we aligned the extracted 16*S* rRNA genes from each genome assembly using MAFFT (107). A maximum likelihood phylogeny was built based on the 16*S* rRNA genes using the program RAxML v.8.2.11 (108) with

a general time reversible (GTR) nucleotide substitution model (109) and 1,000 bootstrap replicates. We used the Interactive Tree of Life (iToL v4) (110) to visualize the phylogenetic tree. The labels of each branch are color-coded by class level. A strip data set represents the genome size of each strain, from light gray for small sizes to black for large genomes. A multivalue bar chart placed around the tree displays the total BGC amount of each strain, in parallel highlighting the specific amount of NRPS, PKS, and the hybrid BGCs of both types. Partial BGCs on contigs <10 kb of WGS projects are not included in this analysis.

The phylogenetic tree of the genus *Chitinophaga* is based on a clustal W alignment (111) of available 16*S* rRNA gene sequences of 25 *Chitinophaga* strains available for cultivation. The tree was calculated using MEGA v7.0.26 with the maximum-likelihood method and GTR-Gamma model (112). Percentage on the tree branches indicate values of 1,000 bootstrap replicates with a bootstrap support of more than 50%. The tree is drawn to scale, with branch lengths measured in the number of substitutions per site.

**BiG-SCAPE-CORASON analysis.** Individual .gb files were processed using antiSMASH 5.0, including the ClusterFinder border prediction algorithm to automatically trim BGCs where gene cluster borders were possible to predict (23, 50). The "hybrids" mode of BiG-SCAPE v1.0.0 (26), which allows BGCs with mixed annotations to be analyzed together, was enabled. Several cutoffs of 0.1 to 0.9 were tested. After manual inspection, the network with a cutoff of 0.6 was visualized and annotated using Cytoscape v3.6.0 (113).

**Mass spectrometric analysis.** A quadrupole time-of-flight spectrometer (LC-QTOF maXis II, Bruker Daltonik) equipped with an electrospray ionization source in line with an Agilent 1290 infinity LC system (Agilent) was used for all UHPLC-QTOF-HR-MS and MS/MS measurements. C18 RP-UHPLC (ACQUITY UPLC BEH C18 column [130 Å, 1.7 $\mu$m, 2.1 × 100 mm]) was performed at 45°C with the following linear gradient: 0 min: 95% A; 0.30 min: 95% A; 18.00 min: 4.75% A; 18.10 min: 0% A; 22.50 min: 0% A; 22.60 min: 95% A; 25.00 min: 95% A (A: H$_2$O, 0.1% HCOOH; B: CH$_3$CN, 0.1% HCOOH; flow rate: 0.6 mL/min). Mass spectral data were acquired using a 50 to 2,000 *m/z* scan range at 1 Hz scan rate. MS/MS experiments were performed with 6 Hz and the top five most intense ions in each full MS spectrum were targeted for fragmentation by higher-energy collisional dissociation at 25 eV or 55 eV using N$_2$ at 10$^{-2}$ mbar. Precursors were excluded after two spectra, released after 0.5 min and reconsidered if the intensity of an excluded precursor increased by factor 1.5 or more.

**Chemotype-barcoding matrix. (i) Raw data processing.** Raw data processing was performed with DataAnalysis 4.4 (Bruker) using recalibration with sodium formate. RecalculateLinespectra with two thresholds (10,000 and 20,000) and subsequent FindMolecularFeatures (0.5 to 25 min, S/N = 0, minimal compound length = 8 spectra, smoothing width = 2, correlation coefficient threshold = 0.7) was performed. Bucketing was performed using ProfileAnalysis 2.3 (Bruker; 30 to 1,080 s, 100 to 6,000 *m/z*, Advanced Bucketing with 24 s 5 ppm, no transformation, Bucketing basis = H$^+$).

**(ii) Data management.** The complete data set was copied and processed with both line spectra thresholds (10,000 and 20,000). Bucketing was performed on both sets at the same time, resulting in one table containing all buckets deemed identical for every sample (under both thresholds). This table was subsequently curated: Buckets were only deemed present, if they were detected in the 20,000 samples (4,707 buckets did not meet this criteria and were deleted). To avoid "uniqueness" due to compounds being just above the detection threshold, entries from the 10,000 set were used for these buckets, the 20,000 entries were deleted. Eight buckets were not filled in this table and were subsequently deleted, resulting in a final table of 278 samples with 4,188 buckets.

For preparation of the barcode, for each strain (and combined control) and fermentation length (4 and 7 days) how often each bucket was present was analyzed. Because media controls were repeated several times (instead of once per strain/duration combination), they were combined first. Each bucket present in one of the control/time/media combinations was deemed present.

Classifications were "more than one" for buckets present in two or more media; "media X" for buckets present in only one media and "none" for all not present at that strain/duration combination. This table is the input table for the barcode matrix. For each strain, it was calculated if a bucket was present in at least one of its media/duration combinations. Subsequently, how many strains (excluding media controls) each bucket was present was calculated.

To achieve a readable representation, the buckets in the input table were sorted (after transformation/rotation) (i) (alphabetically) for each sample (for defragmentation of media), (ii) (decreasing) for the number of strains in which the bucket is present, and (iii) presence in any of the control conditions. Visualization was performed with the R-script deposited in the SI. All MS and MS/MS-raw data and used data sets are deposited at http://dx.doi.org/10.24406/fordatis/188.

**Cultivation and screening conditions. (i) Bacterial strains and culture conditions.** All 25 *Chitinophaga* strains used for this study (Table S1) were purchased from Deutsche Sammlung von Mikroorganismen und Zellkulturen (DMSZ) and Korean Collection for Type Cultures (KCTC). Strains were inoculated in 50 mL R2A (114) (HiMedia Laboratories, LLC.) in 300-mL flasks and incubated for 3 days, then 2% (vol/vol) culture volume was used to inoculate the main cultures (50 mL media/300 mL flask), incubated at 28°C with agitation at 180 rpm for 4 or 7 days. R2A, 3018 (1 g/L yeast extract, 5 g/L Casitone, pH 7.0, 24 mM *N-acetylglucosamine* added after autoclaving), 5294 (10 g/L soluble starch, 10 g/L glucose, 10 g/L glycerol 99%, 2.5 g/L liquid corn steep, 5 g/L peptone, 2 g/L yeast extract, 1 g/L NaCl, 3 g/L CaCO3, pH 7.2), 3021 (10 g/L glucose, 10 g/L chitin, 5 g/L soy flour, 5 g/L casein peptone, pH 7.0), and 5065 (15 g/L soluble starch, 10 g/L glucose, 10 g/L soy flour, 1 g/L yeast extract, 0.1 g/L K2HPO4, 3 g/L NaCl, pH 7.4) supplemented with 1 mL/L SL-10 trace element solution (115) and 3 mL/L vitamin solution (added after autoclaving, VL-55 medium, DSMZ) were used as main culture media.

**(ii) Screening conditions.** Freeze-dried cultures were extracted with 40-mL MeOH, dried and resuspended in 1-mL MeOH to generate 50x concentrated extracts, partially used for extract analysis by UHPLC-HR-MS (details in MS analysis part). An aliquot of 150 $\mu$L of each extract was dried again and

resuspended in 75 $\mu$L DMSO, resulting in 100x concentrated extracts. Four concentrations (1x, 0.5x, and 0.25x twice) of each extract were screened in 384-well plate format (20 $\mu$L assay volume) supported by liquid handling robots (Analytik Jena CyBio Well 96/384 CYBIO SW-CYBI, Thermo Fisher Scientific Matrix Wellmate and Multidrop). Assay plates were inoculated with 20,000 CFU/mL from overnight precultures (100-mL flasks filled with 30-mL cation adjusted Mueller-Hinton II Broth [MHIIB, BD Difco], 37°C) diluted in MHIIB of *Escherichia coli* ATCC 35218, *E. coli* ATCC 25922 ΔTolC, *Pseudomonas aeruginosa* ATCC 27853, *Klebsiella pneumoniae* ATCC 13883, or *Staphylococcus aureus* ATCC 25923 and incubated at 37°C over-night. The protocol was adapted to an inoculum of 100,000 CFU/mL for *Mycobacterium smegmatis* ATCC 607, *Candida albicans* FH2173, and 50,000 spores/mL from a spore solution for *Aspergillus flavus* ATCC 9170. Brain heart infusion (BHI) broth supplemented with 1% (vol/vol) Tween 80 for *M. smegmatis* and MHIIB for *A. flavus* and *C. albicans* were used. Pre-cultures and main-cultures were incubated at 37°C for 2 days, except the pre-culture of *C. albicans* was incubated at 28°C. All assay plates and pre-cultures were incubated with agitation at 180 rpm and controlled humidity (80% RH) to prevent evaporation. Active extracts were defined as >80% growth inhibition compared with the controls. Optical density (600 nm) or for *A. flavus*, *C. albicans*, and *M. smegmatis* a quantitative ATP assay (BacTiter-Glo, Promega) based on the resulting relative light units, both measured with a LUMIstar OPTIMA Microplate Luminometer (BMG LABTECH), was used according to the manufacture's protocol to detect growth inhibition. UHPLC-HR-ESI-MS guided fractionation helped to identify and dereplicate possible active molecules within crude extracts.

**Strain fermentation and purification of chitinopeptins.** *C. eiseniae* DSM 22224 and *C. flava* KCTC 62435 were inoculated from plate (R2A) in 300-mL Erlenmeyer flasks filled with 100 mL R2A and incubated at 28°C with agitation at 180 rpm for 3 days. Followed by 20 L fermentations (separated in 500 mL culture volume per 2 L flasks) in 3018-medium inoculated with 2% (vol/vol) pre-culture, incubated under the same conditions for 4 days, and subsequently freeze-dried. Dried cultures were extracted with one-time culture volume MeOH. The extracts were evaporated to dryness using rotary evaporation under reduced pressure, resuspened in 3 L of 10% MeOH/$H_2O$, and loaded onto a XAD16N column (1 L bed volume). Step-wise elution with 10%, 40%, 60%, 80%, and 100% MeOH (two-times bed volume each) was performed. The 80% and 100% fractions containing chitinopeptins were subjected to preparative followed by semi-preparative reverse-phase high-performance liquid chromatography (RP-HPLC) on C18 columns (preparative HPLC: Synergi 4 $\mu$m Fusion-RP 80 Å [250 × 21.2 mm]; semi-preparative-HPLC: Synergi 4 $\mu$m Fusion-RP 80 Å [250 × 10 mm], mobile phase: $CH_3CN/H_2O$ + 0.1% HCOOH) using linear gradients of 5% to 95% organic in 40 min. Final purification was achieved using UHPLC on a ACQUITY UPLC BEH C18 column (130 Å, 1.7 $\mu$m, 100 × 2.1 mm) with the same mobile phase and a linear gradient of 30% to 60% organic in 18 min. After each step, fractions containing chitinopeptins were evaporated to dryness using a high performance evaporator (Genevac HT-12).

Chitinopeptin A (1). White, amorphous powder; $[\alpha]_D^{20}$ = −27.6° (*c* = 0.11, $CH_3OH$); LC-UV ($CH_3CN$ in $H_2O$ + 0.1% HCOOH) $\lambda$max 220 nm; [1H-NMR] and [13C]-NMR data, see Table S2 and S3; HR-MS (ESI-TOF) *m/z* (M+H)$^+$ calcd for $C_{82}H_{138}N_{17}O_{28}$ 1809.0037, found 1809.0052.

Chitinopeptin B (2). White, amorphous powder; $[\alpha]_D^{20}$ = -21.0° (*c* = 0.16, $CH_3OH$); LC-UV [($CH_3CN$ in $H_2O$ + 0.1% HCOOH)] $\lambda$max 222 nm; [1H-NMR] and [13C]-NMR data, see Table S2 and S3; HR-MS (ESI-TOF) *m/z* [M+H]$^+$ calcd for $C_{81}H_{136}N_{17}O_{28}$ 1794.9880, found 1794.9907.

Chitinopeptin C1+C2 (3 and 4). Slightly yellowish, amorphous powder; LC-UV [($CH_3CN$ in $H_2O$ + 0.1% HCOOH)] $\lambda$max 219 nm; [1H-NMR] and [13C]-NMR data, see Table S2 and S3; HR-MS (ESI-TOF) *m/z* [M+H]$^+$ calcd for $C_{84}H_{141}N_{18}O_{30}$ 1882.0061, found 1882.0177.

Chitinopeptin D1+D2 (5 and 6). Slightly yellowish, amorphous powder; LC-UV [($CH_3CN$ in $H_2O$ + 0.1% HCOOH)] $\lambda$max 221 nm; [1H-NMR] and [13C]-NMR data, see Table S2 and S3; HR-MS (ESI-TOF) *m/z* [M+H]$^+$ calcd for $C_{83}H_{139}N_{18}O_{30}$ 1867.9905, found 1868.0059.

**Structure elucidation. (i) NMR studies.** NMR spectra of chitinopeptin A (1) were acquired on a Bruker AVANCE 700 spectrometer operating at a proton frequency of 700.13 MHz and a $^{13}$C-carbon frequency of 176.05 MHz. NMR spectra of the remaining compounds were recorded on a Bruker AVANCE 500 spectrometer operating at a proton frequency of 500.30 MHz and a $^{13}$C-carbon frequency of 125.82 MHz. Both instruments were equipped with a 5 mm TCI cryo probe. For structure elucidation and assignment of proton and carbon resonances 1D-$^1$H, 1D-$^{13}$C, DQF-COSY, TOCSY (mixing time 80 ms), ROESY (mixing time 150 ms), multiplicity edited-HSQC, and HMBC spectra were acquired. The raw data is deposited at http://dx.doi.org/10.24406/fordatis/188.

Homonuclear experiments (1D-$^1$H, DQF-COSY, TOCSY, ROESY) were acquired in a mixture of $H_2O$ and $CD_3CN$ in a ratio of 1:1. 1D-$^{13}$C, HSQC, and HMBC spectra were acquired in a mixture of $D_2O$ and $CD_3CN$ in a ratio of 1:1. In the case of CLP 1 the HMBC spectrum has also been acquired in $H_2O/CD_3CN$. $^1$H-chemical shifts were referenced to sodium-3-(Trimethylsilyl)propionate-2,2,3,3-d$_4$. $^{13}$C-chemical shifts were referenced to the solvent signal ($CD_3CN$, $^{13}$C: 1.30 ppm).

**(ii) Advanced Marfey's analysis.** The absolute configuration of all amino acids was determined by derivatization using Marfey's reagent (72). Stock solutions of amino acid standards (50 mM in $H_2O$), NaHCO$_3$ (1 M in $H_2O$), and $N_\alpha$-(2,4-dinitro-5-fluorophenyl)-L-valinamide (L-FDVA, 70 mM in acetone) were prepared. Commercially available and synthesized standards were derivatized using molar ratios of amino acid to FDVA and NaHCO$_3$ (1/1.4/8). After stirring at 40°C for 3 h, 1 M HCl was added to obtain concentration of 170 mM to end the reaction. Samples were subsequently evaporated to dryness and dissolved in DMSO (final concentration 50 mM). L- and D-amino acids were analyzed separately using C18 RP-UHPLC-MS (A: $H_2O$, 0.1% HCOOH; B: $CH_3CN$, 0.1% HCOOH; flow rate: 0.6 mL/min). A linear gradient of 5% to 75% B in 35 min was applied to separate all commercially available amino acid standards except of D- and D-allo-Ile. Separation of D- and D-allo-Ile was archived using chiral HPLC (CHIRALPAK IC, 1 mL/min, 75% n-hexane, 25% isopropyl alcohol, 0.2% HCOOH). Prepared Marfey's adducts of synthesized

$\beta$-hydroxyaspartic acids were analyzed using C18 RP-UHPLC-MS with a linear gradient of 5% to 20% B in 18 min. For the determination of stereochemistry of $\beta$-hydroxyphenylalanines and $\beta$-hydroxyisoleucines, Cbz-protected intermediates were directly subjected to acidic hydrolysis (6 M HCl, 2h, 120°C) and subsequent treatment with Marfey's reagent on analytical scale as described before after drying. Marfey's adducts were analyzed using C18 RP-UHPLC-MS with 5% to 40% B in 18 min.

Total hydrolysis of the chitinopeptins A to D was carried out by dissolving 250 $\mu$g of each peptide in 6 M DCl in $D_2O$ and stirring for 7 h at 160°C. The samples were subsequently evaporated to dryness. Samples were dissolved in 100 $\mu$L $H_2O$, derivatized with l-FDVA and analyzed using the same parameters as described before.

**(iii) Synthesis of $\beta$-hydroxyaspartic acids, $\beta$-hydroxyphenylalanines, and $\beta$-hydroxyisoleucines.** The syntheses of the $\beta$-hydroxyamino acids were achieved by modified literature known procedures. The enantiomeric excess of all synthesized amino acids were determined by chiral HPLC on selected intermediates. The synthesized compounds were fully characterized and/or compared with literature known references. For details of syntheses and characterization see supplemental materials.

**Optical rotation.** Specific rotation was determined on a digital polarimeter (P3000, A. Krüss Optronic GmbH, Germany). Standard wavelength was the sodium d-line with 589 nm. Temperature, concentration, (g/100 mL) and solvent are reported with the determined value.

**MIC.** The MIC was determined by broth microdilution method in 96-well plates with 100 $\mu$L assay volume per well following EUCAST instructions (116, 117) for *E. coli* ATCC 35218, *E. coli* ATCC 25922 ΔTolC, *E. coli* MG1655, *Pseudomonas aeruginosa* ATCC 27853, *P. aeruginosa* PAO750, *Klebsiella pneumoniae* ATCC 13883, *Moraxella catarrhalis* ATCC 25238, *Acinetobacter baumannii* ATCC 19606, *Bacillus subtilis* DSM 10, *Staphylococcus aureus* ATCC 25923, and *Micrococcus luteus* DSM 20030. *Listeria monocytogenes* DSM 20600 (1 day of incubation at 37°C) and *Mycobacterium smegmatis* ATCC 607 (2 days of incubation at 37°C) were grown in BHI broth supplemented with 1% (vol/vol) Tween 80. *Candida albicans* FH2173 was grown in MHIIB over 2 days of incubation at 37°C (pre-culture 2 days at 28°C). Approximate inoculation cell density for all strains was 500,000 CFU/mL, except 1,000,000 CFU/mL were used for *M. smegmatis* and *C. albicans*. For all fungal screenings 100,000 spores/mL were used and incubated at 37°C for *Aspergillus flavus* ATCC 9170 (MHIIB) and 25°C for *Z. tritici* MUCL45407 (4 g/L yeast extract, 4 g/L malt extract, 4 g/L sucrose) and *Fusarium oxysporum* ATCC 7601 (potato dextrose broth, Sigma). Growth inhibition was detected by optical density (600 nm) or for *M. luteus*, *L. monocytogenes*, *C. albicans*, *M. smegmatis*, and all fungi by quantification of free ATP using BacTiter-Glo after 24 to 48 h (72 h for *Z. tritici*). All MIC assays were performed at least in triplicate. FE (III)-citrate was added at molar ratios of 1:1 to samples in DMSO to test compounds in their iron bound conformation.

**Data availability.** MS, MS/MS, and NMR data are deposited at Fordatis - Research Data Repository of Fraunhofer-Gesellschaft (http://dx.doi.org/10.24406/fordatis/188).

## SUPPLEMENTAL MATERIAL

Supplemental material is available online only.

**SUPPLEMENTAL FILE 1**, PDF file, 7.7 MB.

**SUPPLEMENTAL FILE 2**, XLSX file, 0.1 MB.

## ACKNOWLEDGMENTS

We intend to show a deep sense of gratitude to Sanja Mihajlovic as caretaker of the biobank. We also thank Mona-Katharina Bill, Sandra Semmler, Sören M. M. Schuler, Frank Förster, Christine Wehr, Jennifer Kuhn, Nadine Zucchetto, Sina Serife Abdo, Regina Zweigert, Mona Abdullahi, and Kirsten-Susann Bommersheim for their support and valuable discussions as well as Heiko Heese and Joachim Kluge for HPLC purifications of synthetic intermediates and Karin Rahn-Hotze for ee-determininations. We thank Stefan Bernhardt (Institute for Organic Chemistry, Justus-Liebig-University Giessen) for chiral HPLC analysis. This work was financially supported by the Hessen State Ministry of Higher Education, Research and the Arts (HMWK) via the state initiative for the development of scientific and economic excellence for the LOEWE Center for Insect Biotechnology and Bioresources. Sanofi-Aventis Deutschland GmbH and Evotec International GmbH funded this work in the framework of the Sanofi-Fraunhofer Natural Products Center and its follow up, the Fraunhofer-Evotec Natural Products Center of Excellence.

S.B. and M.S. conceived and designed the experiments. S.B., M.S., M.A.P., B.L., M.M., C.H., A.Bi., Y.K., C.P., and M.K. performed the experiments. S.B., M.S., M.A.P., C.P., Y.K., A.Ba., L.T., M.K., and T.F.S. analyzed the data. A.Ba., L.T., J.G., and P.E.H. initiated the project idea. J.G., M.S., and T.F.S. supervised the project. A.V. and P.E.H. initiated the public-private partnership between Fraunhofer and Sanofi (later Evotec). S.B., M.S., and T.F.S. drafted the first manuscript. S.B., M.S., and T.F.S. revised the manuscript. All authors accepted the final version of the manuscript.

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
