## [Reviewer comments · Microbiology Spectrum]

Microbiology Spectrum

Genomic and chemical decryption of the Bacteroidetes phylum for its potential to biosynthesize natural products

Stephan Brinkmann, Michael Kurz, Maria Patras, Christoph Hartwig, Michael Marnier, Benedikt Leis, André Billion, Yolanda Kleiner, Armin Bauer, Luigi Toti, Christoph Pöverlein, Peter Hammann, Andreas Vilcinskis, Jens Glaeser, Marius Spohn, and Till Schäberle

Corresponding Author(s): Till Schäberle, University of Giessen

Review Timeline:

Submission Date:	December 2, 2021
Editorial Decision:	February 11, 2022
Revision Received:	March 26, 2022
Accepted:	March 29, 2022

Editor: Eva Sonnenschein

Reviewer(s): Disclosure of reviewer identity is with reference to reviewer comments included in decision letter(s). The following individuals involved in review of your submission have agreed to reveal their identity: Scott Alexander Jarmusch (Reviewer #1); John Vollmers (Reviewer #2)

Transaction Report:

DOI: <https://doi.org/10.1128/spectrum.02479-21>

February 11, 2022

Prof. Till F. Schäberle
University of Giessen
Institute for Insect Biotechnology
Ohlebergsweg 12
Giessen 35392
Germany

Re: Spectrum02479-21 (Genomic and chemical decryption of the Bacteroidetes phylum for its potential to biosynthesize natural products)

Dear Prof. Till F. Schäberle:

As indicated by the reviewer, please deposit your chemical data in a repository and thereby make it accessible to the community; e.g. at Metabolomics Workbench or GNPS-MassIVE.

Link Not Available

Sincerely,

Eva Sonnenschein

Journals Department
Reviewer comments:

Reviewer #1 (Comments for the Author):

General Comments

The work by Brinkmann et al. examines the biosynthetic potential of the underexplored phylum Bacteroidetes, with specific

focus on the chemical potential of what they pin down as a talented genera, Chitinophaga. Via a thorough biosynthetic analysis, the authors utilize publicly available genomes to thoroughly characterize numerous strains that have not been looked at in terms of natural products work. Focusing in on one of the more talented genera, Chitinophaga, the authors conduct OSMAC to cover as much of the metabolic potential of the strains. Bioassay-guided fractionation lead to the discovery of 6 new metabolites, cyclic lipopeptides, that represent the first CLPs isolated from this genera. Their biological activity falls in line with other similar cyclic lipopeptides.

I think the work as really solid foundations and doesn't require addition experiments. It is a well thought out piece of work that covers the broad range of topics, from biosynthetic potential to isolation of new metabolites with biological activities. Some conclusions go too far I think. One main one is the novelty of the CLPs discovered. CLPs have been known since the golden age of antibiotics, the 1950s (Baltz 2021 - Genome mining for drug discovery: cyclic lipopeptides related to daptomycin | Journal of Industrial Microbiology and Biotechnology | Oxford Academic (oup.com)), and their biological activity is well understood as well as being some of the most studied metabolites regarding the big three mentioned in this paper: Streptomyces, Bacillus and Pseudomonas. New CLPs are great results but I do not think they represent novel chemical space (in terms of the larger context of natural products), new chemical space for sure but not novel. Furthermore, the some of the challenges of CLPs (and macrocyclic peptides in general) as potential drug candidates has little to do with the discovery of new metabolites (which is fairly common) but multiple other factors laid out by Vinogradov et al. 2019 (Macrocyclic Peptides as Drug Candidates: Recent Progress and Remaining Challenges | Journal of the American Chemical Society (acs.org)).

The researchers here have filled a significant gap in the literature and work like this serves as an important primer for future studies but also to inform other researchers of the potential in a group of understudied taxa. The LCMS data (and the NMR data as well) definitely needs to be deposited into a repository so the community can benefit from all of the hard work you did, maybe consider Metabolomics Workbench or GNPS-MassIVE.

Detailed Comments

Line 72 - Underutilized is completely incorrect. I think the weight of the literature regarding new CLPs discovered annually and their continued investigation in many type strains renders this term incorrect. They are a very well-known group of metabolites, therefore, this terminology needs to be changed.

Line 88 - changed plethora to 'the vast majority'.

Line 111 & 225 - non-connatural is strange jargon. Simplify this maybe to something like, 'unique compared to....'

Line 116 - This is an expected results since all databases lack data from these genera. This represents a great in road for future research trying the same thing.

Line 162 - remove remarkable (generally remove hyperbole)

Line 243 - Need a reference to OSMAC - Bode et al. 2002 Big Effects from Small Changes: Possible Ways to Explore Nature's Chemical Diversity - Bode - 2002 - ChemBioChem - Wiley Online Library

Line 248 - How are you defining UHR? QToF are typically not ultra high resolution instruments, only FTICR-MS usually has this distinction. QToF are ~50K resolution, Orbitraps are 50K-1 million and FTICR is 1 million+. I recommend you change all instances of UHR to HR.

Lines 267-290. This needs to be expanded upon in the discussion! You make a great point regarding culturing times which I believe makes some in the community to believe bacteria (like gut microbes) are not biosynthetically worthwhile investigating due to sort culturing times (24 hours). You need to cite the perspective Bill Gerwick published regarding culturing times. J. Antibiotics, 2020, 73, 481-87.

Lines 297-302. This sections shows two things: 1) NP libraries aren't fit for dereplication of understudied taxa and 2) this is why data needs to be deposited from studies such as this.

Lines 327. I don't see anything specific (like an addition ring closure) that should cause extra stability in these CLPs. I do not think you're working in the 'elevated CE range' either. 20 eV (please correct me if I read the supplementary data wrong - it is tough to read) is not high energy. 40 eV should induce reasonable fragmentation that would have enabled molecular networking.

Line 375. Eliminate this line, it is irrelevant for the study.

Lines 398-403. Great job here. I am disappointed that Fig. S38 is not represented as concentration since you had pure metabolites to test against (generate a cal curve). I think what you may have stumbled upon is that CLPs can act as siderophores but it is only one of the functions of these secondary metabolites. Clearly they are antimicrobial, iron chelating, and play larger roles in things like bacterial swarming (as seen in Bacilli and Pseudomonads). Maybe worth briefly mentioning the large ecological context.

Line 473- remove 'source' and replace with 'mine'

Line 475 - use metabolites instead of compounds

Line 478 - use sequentially instead of 'in sequel' and place 'have' between technologies and already.

Line 483-488 - possibly cite new review in NPR by Jarmusch et al. 2021 regarding Big data and MS-based approaches (Advancements in capturing and mining mass spectrometry data are transforming natural products research - Natural Product Reports (RSC Publishing))

Line 496 - some actinobacteria. It doesn't get close to Streptomyces or Amycolatopsis.

Figure 1 - Very nice figure! Readability of the colors is very poor. All of the dark colored nodes in (b) look black. Also for color blind readers it is very difficult (speaking personally). You could use iWantHue (medialab.github.io) as a resource for making colored figures.

Figure 3a. Axis and phylogenetic tree distances are impossible to interpret. Are we looking m/z on the x axis of the barcoding?

Figure 4. I'm fine with the R/S stereochemistry but I think labelling these with D/L nomenclature makes it more usable for the broad community outside of chemists. Also makes it easier to tie to the biosynthetic genes.

Line XXX - You have discovered novel chemistry but I think you are over selling the work you have done here. CLPs would be considered 'low hanging fruit' since biosynthetically they are so easy to predict.

Reviewer #2 (Comments for the Author):

The overall approach seems straightforward and solid. However, I do have a few moderate to minor issues:

MODERATE ISSUES:

Lines 137 & 543 Please define what exactly is meant with "correctly annotated". Did this involve manual curation of the annotated datasets? In that case please state how you would recognize incorrectly annotated datasets

Lines 148-150 & lines 171-173: In lines 148-150 the authors first state that some classes display less significance for NP discovery than others, due to the lower number of BGC detected in the corresponding genomes. However in lines 171-173 they state that the simple number of BGCs detected is less important than their novelty (or "divergence" as the authors put it), a statement to which I whole-heartedly agree, but which strongly contradicts the first point. Wouldn't, for example, a taxon that yields a relatively low number but nonetheless high diversity of BGCs that are unique for this taxon be of actually more significance for novel NP discovery than taxa rich in BGCs that show close homologs in various other taxa? This should be better reflected in the wording.

Line 187: I think "other PKS" should also be placed in quotation marks

Lines 220-223: It would appear to me, that any sequence based screening for such broad functional categories as "secondary metabolite production" is likely to yield also some spurious and/or weakly supported results. Given also the large number of hypothetical proteins that are to be expected in the respective pan-genome of each phylum, many of such hits will be without any useful functional annotation and would therefore not appear in the MiBIG database, even if close homologs could be found in different phyla. Therefore, even if it might be true, I don't think the here presented analyses here actually prove a "tremendous uniqueness" of biosynthetic gene clusters within the Bacteroidetes. The authors should consider this and comment on it within the manuscript.

Lines 542-544: Please define "WGS projects of strains of interest". Clearly this means non-circularised draft genomes, but since there is a huge number available, was that selection based on exactly (e.g. specific N50/N80-cutoffs?)

Line 544: Supplementary table S8 does not seem to be available?

Line 288: "This high level of strain-specific metabolites": Could you provide some reference numbers here, in order for the reader to evaluate if this is actually a high number compared to other taxa?

Line 297-302: considering that the number of representative metabolites determined for the chitinophaga set alone (2736) is already significantly higher than the entire in-house reference database (1700), how representative is this comparison? Is it even possible to represent an equivalent range of metabolites in this database (especially considering the authors also show earlier that the range of metabolites captured is strongly dependent on the exact growth conditions)?

Lines 411-414: Is data shown for this anywhere in the manuscript or supplemental ? please reference.

MINOR ISSUES:

Lines 108 and 110-112: first mentioning that the NP production is limited to a few taxonomic hotspots within the Bacteroidetes, rather than the whole phylum but then announcing an "overall high potential to discover novel scaffolds from this phylum" may easily appear contradictory. The authors are actually distinguishing between the sheer NUMBER of potential NP biosynthesis gene clusters concentrated mostly in a few Bacteroidetes taxa and the apparent NOVELTY of said clusters, which appears to be relatively high for the entire Bacteroidetes phylum. In order to avoid misunderstanding, the authors may want to rephrase that section slightly in order to make that more clear.

Lines 128-129: "The large and diverse phylum Bacteroidetes harbors Gram-stain-negative, chemo-organotrophic, 128 non-spore forming rod shaped bacteria, graded into six so-called classes. " I think some literature should be cited also for these general properties of the phylum.

Line 141-143. This kind of figure description belongs into the respective figure legend, not into the main text.

Line 185. I think a more precise wording would be "...only 12 GCFs clustered with MiBIG reference BGCs of known function"?

Line 401: "3018 medium" does not seem to be commonly familiar? please provide a reference.

Line 404: why directly specify the exact number of fungi, but not of gram positive and gram negative bacteria here? It would appear better to specify "[...] tested against 8 gram-negative and 5 gram-positive bacterial strains, as well as [...]"

Line 511: "[...] is in similar range than [...]". Wording seems strange. Maybe change to "shows a similar range compared to [...]"?

ADDITIONAL NOTES ON WORDING:

Abstract line 39: perhaps drop the "the" in "we isolated THE new iron chelating[...]?"

Lines 95&96: "[...] towards a to date yet underexplored [...]" wording seems a bit off. Maybe change to "[...] toward a currently still underexplored [...]"?

Line 144-145: "[...]enabled classes and genera comparison in terms of [...]" seems like strange wording. Maybe "[...] enabled comparisons between different classes and genera in terms of [...]" would be more correct?

Line 168: The term "second best genus" seems oddly subjective... Maybe a better wording would be "The genus with the second highest BGC load within the Bacteroidetes phylum was Taibaiella [...]"?

Line 277: "..were detected after seven then after four ..." correct "then" to "than"

Line 288: over the course of the manuscript it can be a little hard to follow which reference sets were uses for which analyses, therefore the wording "[...] 10 out of 25 [...]" may seem a little confusing. It may be better to specify here again: "[...] 10 out of the 25 total analyzed strains [...]"

Staff Comments:

Preparing Revision Guidelines

Please return the manuscript within 60 days; if you cannot complete the modification within this time period, please contact me. If you do not wish to modify the manuscript and prefer to submit it to another journal, please notify me of your decision immediately so that the manuscript may be formally withdrawn from consideration by Microbiology Spectrum.

General Comments

The work by Brinkmann et al. examines the biosynthetic potential of the underexplored phylum Bacteroidetes, with specific focus on the chemical potential of what they pin down as a talented genera, *Chitinophaga*. Via a thorough biosynthetic analysis, the authors utilize publically available genomes to thoroughly characterize numerous strains that have not been looked at in terms of natural products work. Focusing in on one of the more talented genera, *Chitinophaga*, the authors conduct OSMAC to cover as much of the metabolic potential of the strains. Bioassay-guided fractionation lead to the discovery of 6 new metabolites, cyclic lipopeptides, that represent the first CLPs isolated from this genera. Their biological activity falls in line with other similar cyclic lipopeptides.

I think the work as really solid foundations and doesn't require addition experiments. It is a well thought out piece of work that covers the broad range of topics, from biosynthetic potential to isolation of new metabolites with biological activities. Some conclusions go too far I think. One main one is the novelty of the CLPs discovered. CLPs have been known since the golden age of antibiotics, the 1950s (Baltz 2021 - Genome mining for drug discovery: cyclic lipopeptides related to daptomycin | Journal of Industrial Microbiology and Biotechnology | Oxford Academic (oup.com)), and their biological activity is well understood as well as being some of the most studied metabolites regarding the big three mentioned in this paper: Streptomyces, Bacillus and Pseudomonas. New CLPs are great results but I do not think they represent novel chemical space (in terms of the larger context of natural products), new chemical space for sure but not novel. Furthermore, the some of the challenges of CLPs (and macrocyclic peptides in general) as potential drug candidates has little to do with the discovery of new metabolites (which is fairly common) but multiple other factors laid out by Vinogradov et al. 2019 (Macrocyclic Peptides as Drug Candidates: Recent Progress and Remaining Challenges | Journal of the American Chemical Society (acs.org)).

The researchers here have filled a significant gap in the literature and work like this serves as an important primer for future studies but also to inform other researchers of the potential in a group of understudied taxa. The LCMS data (and the NMR data as well) definitely needs to be deposited into a repository so the community can benefit from all of the hard work you did, maybe consider Metabolomics Workbench or GNPS-MassIVE.

Detailed Comments

Line 72 – Underutilized is completely incorrect. I think the weight of the literature regarding new CLPs discovered annually and their continued investigation in many type strains renders this term incorrect. They are a very well-known group of metabolites, therefore, this terminology needs to be changed.

Line 88 – changed plethora to 'the vast majority'.

Line 111 & 225 – non-connatural is strange jargon. Simplify this maybe to something like, 'unique compared to....'

Line 116 – This is an expected results since all databases lack data from these genera. This represents a great in road for future research trying the same thing.

Line 162 – remove remarkable (generally remove hyperbole)

Line 243 – Need a reference to OSMAC – Bode et al. 2002 Big Effects from Small Changes: Possible Ways to Explore Nature's Chemical Diversity - Bode - 2002 - ChemBioChem - Wiley Online Library

Line 248 – How are you defining UHR? QToF's are typically not ultra high resolution instruments, only FTICR-MS usually has this distinction. QToF's are ~50K resolution, Orbitraps are 50K-1 million and FTICR is 1 million+. I recommend you change all instances of UHR to HR.

Lines 267-290. This needs to be expanded upon in the discussion! You make a great point regarding culturing times which I believe makes some in the community to believe bacteria (like gut microbes) are not biosynthetically worthwhile investigating due to sort culturing times (24 hours). You need to cite the perspective Bill Gerwick published regarding culturing times. *J. Antibiotics*, 2020, 73, 481-87.

Lines 297-302. This sections shows two things: 1) NP libraries aren't fit for dereplication of understudied taxa and 2) this is why data needs to be deposited from studies such as this.

Lines 327. I don't see anything specific (like an addition ring closure) that should cause extra stability in these CLPs. I do not think you're working in the 'elevated CE range' either. 20 eV (please correct me if I read the supplementary data wrong – it is tough to read) is not high energy. 40 eV should induce reasonable fragmentation that would have enabled molecular networking.

Line 375. Eliminate this line, it is irrelevant for the study.

Lines 398-403. Great job here. I am disappointed that Fig. S38 is not represented as concentration since you had pure metabolites to test against (generate a cal curve). I think what you may have stumbled upon is that CLPs can act as siderophores but it is only one of the functions of these secondary metabolites. Clearly they are antimicrobial, iron chelating, and play larger roles in things like bacterial swarming (as seen in Bacilli and Pseudomonads). Maybe worth briefly mentioning the large ecological context.

Line 473- remove 'source' and replace with 'mine'

Line 475 – use metabolites instead of compounds

Line 478 – use sequentially instead of 'in sequel' and place 'have' between technologies and already.

Line 483-488 – possibly cite new review in NPR by Jarmusch et al. 2021 regarding Big data and MS-based approaches (Advancements in capturing and mining mass spectrometry data are transforming natural products research - Natural Product Reports (RSC Publishing))

Line 496 – some actinobacteria. It doesn't get close to Streptomyces or Amycolatopsis.

Figure 1 – Very nice figure! Readability of the colors is very poor. All of the dark colored nodes in (b) look black. Also for color blind readers it is very difficult (speaking personally). You could use iWantHue (medialab.github.io) as a resource for making colored figures.

Figure 3a. Axis and phylogenetic tree distances are impossible to interpret. Are we looking m/z on the x axis of the barcoding?

Figure 4. I'm fine with the R/S stereochemistry but I think labelling these with D/L nomenclature makes it more usable for the broad community outside of chemists. Also makes it easier to tie to the biosynthetic genes.

Line XXX – You have discovered novel chemistry but I think you are over selling the work you have done here. CLPs would be considered 'low hanging fruit' since biosynthetically they are so easy to predict.

Final Decision made for Spectrum02479-21 Point-by-point response

Dear Editor,
Dear Reviewers,

We would like to thank all of you for the work and time invested into our manuscript. We were happy to read that the general idea of our manuscript is suitable for publication in Microbiology Spectrum. We addressed all the points raised and would like to provide a point-by-point response here. Our comments are marked in blue.

Reviewer #1 (Comments for the Author):

General Comments

The work by Brinkmann et al. examines the biosynthetic potential of the underexplored phylum Bacteroidetes, with specific focus on the chemical potential of what they pin down as a talented genera, Chitinophaga. Via a thorough biosynthetic analysis, the authors utilize publicly available genomes to thoroughly characterize numerous strains that have not been looked at in terms of natural products work. Focusing in on one of the more talented genera, Chitinophaga, the authors conduct OSMAC to cover as much of the metabolic potential of the strains. Bioassay-guided fractionation lead to the discovery of 6 new metabolites, cyclic lipopeptides, that represent the first CLPs isolated from this genera. Their biological activity falls in line with other similar cyclic lipopeptides.

I think the work as really solid foundations and doesn't require addition experiments. It is a well thought out piece of work that covers the broad range of topics, from biosynthetic potential to isolation of new metabolites with biological activities. Some conclusions go too far I think. One main one is the novelty of the CLPs discovered. CLPs have been known since the golden age of antibiotics, the 1950s (Baltz 2021 - Genome mining for drug discovery: cyclic lipopeptides related to daptomycin | Journal of Industrial Microbiology and Biotechnology | Oxford Academic (oup.com)), and their biological activity is well understood as well as being some of the most studied metabolites regarding the big three mentioned in this paper: Streptomyces, Bacillus and Pseudomonas. New CLPs are great results but I do not think they represent novel chemical space (in terms of the larger context of natural products), new chemical space for sure but not novel. Furthermore, the some of the challenges of CLPs (and macrocyclic peptides in general) as potential drug candidates has little to do with the discovery of new metabolites (which is fairly common) but multiple other factors laid out by Vinogradov et al. 2019 (Macrocyclic Peptides as Drug Candidates: Recent Progress and Remaining Challenges | Journal of the American Chemical Society (acs.org)).

The researchers here have filled a significant gap in the literature and work like this serves as an important primer for future studies but also to inform other researchers of the potential in a group of understudied taxa. The LCMS data (and the NMR data as well) definitely needs to be deposited into a repository so the community can benefit from all of the hard work you did, maybe consider Metabolomics Workbench or GNPS-MassIVE.

Thank you very much for this positive evaluation of our manuscript. Based on your comments, we revised our manuscript (details see below). We also rephrased the statements about the novelty of the here presented CLPs and discussed the challenges of CLPs been developed into drug candidates in the conclusion (we included the review of Vinogradov et al. 2019). As Fraunhofer employees, we are asked to deposit all raw data (LCMS, LCMS/MS and NMR) at our own Fraunhofer repository called FORDATIS

(<https://fordatis.fraunhofer.de/about.jsp?locale=en>). Therefore, we uploaded all raw data (LCMS, LCMS/MS and NMR data) to FORDATIS and added the corresponding DOI-number (<http://dx.doi.org/10.24406/fordatis/188>) to the manuscript. This enables easy findability and access to the data.

Detailed Comments

Line 72 - Underutilized is completely incorrect. I think the weight of the literature regarding new CLPs discovered annually and their continued investigation in many type strains renders this term incorrect. They are a very well-known group of metabolites, therefore, this terminology needs to be changed.

We agree with the reviewer and agree that the wording used was not appropriate. The sentence was changed and reads now as follows: “~~An underutilized as well as~~ promising NP group is represented by the cyclic lipopeptides (CLPs), sharing a common structural core composed of a lipid tail linked to a cyclized oligopeptide.”

Line 88 - changed plethora to 'the vast majority'.

We have changed it.

Line 111 & 225 - non-connatural is strange jargon. Simplify this maybe to something like, 'unique compared to....'

We have changed it accordingly (underlined = new): “The vast majority of Bacteroidetes BGC of the RiPPs, NRPS, PKS, and hybrid NRPS/PKS classes are ~~non-connatural~~ unique compared to BGCs of any other phylum, ...” (line 111) and “~~Expanding-Extension of~~ this similarity network analysis towards taxonomic relations on phylum level showed that there is no connaturalty of Bacteroidetes BGC of the RiPPs, NRPS, PKS, and hybrid NRPS/PKS classes are unique compared to BGCs of any other phylum (Fig. 2B).” (line 225)

Line 116 - This is an expected results since all databases lack data from these genera. This represents a great in road for future research trying the same thing.

Thank you very much; therefore, we wanted to state it clearly in the manuscript.

Line 162 - remove remarkable (generally remove hyperbole)

We have changed it accordingly and generally removed hyperbole throughout the whole manuscript.

Line 243 - Need a reference to OSMAC - Bode et al. 2002 Big Effects from Small Changes: Possible Ways to Explore Nature's Chemical Diversity - Bode - 2002 - ChemBioChem - Wiley Online Library

The OSMAC reference – Bode et al. 2002 – was already added to the sentence before. Now, we also added it to this sentence for clarification.

Line 248 - How are you defining UHR? QToF's are typically not ultra high resolution instruments, only FTICR-MS usually has this distinction. QToF's are ~50K resolution, Orbitraps are 50K-1 million and FTICR is 1 million+. I recommend you change all instances of UHR to HR.

Thank you for this explanation. We fully agree and changed all instances into HR.

Lines 267-290. This needs to be expanded upon in the discussion! You make a great point regarding culturing times which I believe makes some in the community to believe bacteria (like gut microbes) are not biosynthetically worthwhile investigating due to sort culturing times (24 hours). You need to cite the perspective Bill Gerwick published regarding culturing times. *J. Antibiotics*, 2020, 73, 481-87.

We agree with the reviewer. The mentioned perspective (however the page numbers are referring to the editorial from Bill Fenical) is focusing on the isolation of new bacteria and the influence of cultivation time on succeeding in this research field. While we are also active in this field (<https://www.frontiersin.org/articles/10.3389/fmicb.2020.597628/full>) we are aware of the relevance of cultivation time in this context. Matching the scope of our study and the relevance of cultivation time for succeeding in expression of natural product synthesis, we added a paragraph to our discussion.

Lines 297-302. This sections shows two things: 1) NP libraries aren't fit for dereplication of understudied taxa and 2) this is why data needs to be deposited from studies such as this.

Yes, in case the understudied taxa are genetically similar to known taxa and thus carrying closely similar or identical biosynthetic gene cluster as the taxa known and studied, such libraries might help to prove this relation by dereplicating known natural products. A low frequency of rediscovery of natural products just proves the understudied taxa's value of being investigated in terms of searching for new natural products.

Thank you for suggestion to deposit the raw data. We also see a high value in sharing this data with the research community, and deposited it here (the DOI is now mentioned in the manuscript): <http://dx.doi.org/10.24406/fordatis/188>

Lines 327. I don't see anything specific (like an addition ring closure) that should cause extra stability in these CLPs. I do not think you're working in the 'elevated CE range' either. 20 eV (please correct me if I read the supplementary data wrong - it is tough to read) is not high energy. 40 eV should induce reasonable fragmentation that would have enabled molecular networking.

We agree that there are not any specific structural features (like an addition ring closure) that could explain the stability in these CLPs (and indeed, we did not expect that it will be so difficult to get a good fragment signature). However, we did various MS-measurements up to 55 eV without seeing any fragmentation. For clarification, we added the 55 eV information to the sentence as following: "All six native peptides appeared to be highly stable, since only poor yields of fragment ions arose using electrospray ionization source (Fig. S3B), even under elevated collision energy conditions (up to 55 eV)."

Line 375. Eliminate this line, it is irrelevant for the study.

We eliminated this line.

Lines 398-403. Great job here. I am disappointed that Fig. S38 is not represented as concentration since you had pure metabolites to test against (generate a cal curve). I think what you may have stumbled upon is that CLPs can act as siderophores but it is only one of the functions of these secondary metabolites. Clearly they are antimicrobial, iron chelating, and play

larger roles in things like bacterial swarming (as seen in Bacilli and Pseudomonads). Maybe worth briefly mentioning the large ecological context.

We agree that representing as concentration in Fig. S38 using a calibration curve would have had been the best way. However, due to compound shortage we can't deliver the concentrations at this time point. We are sorry for that. Moreover, we also agree with you that the large ecological context should be mentioned. Therefore, we mentioned it in the discussion part as following (underlined = new): "These NRPS-assembled CLPs exhibit primarily activity against *Candida albicans* and were found to coordinate iron. Binding of metal ions is a feature also described for other CLPs such as pseudofactin II, which displays an increased antimicrobial activity upon metal-coordination due to disruption of the cytoplasmic membrane in its chelated state (95). In general, it is known that individual CLPs possess multiple functions such as iron chelation, antimicrobial activity, and interaction in bacterial motility (19)."

Line 473 - remove 'source' and replace with 'mine'

Line 475 - use metabolites instead of compounds

Line 478 - use sequentially instead of 'in sequel' and place 'have' between technologies and already.

All changes mentioned above have been made in the corresponding lines.

Line 483-488 - possibly cite new review in NPR by Jarmusch et al. 2021 regarding Big data and MS-based approaches (Advancements in capturing and mining mass spectrometry data are transforming natural products research - Natural Product Reports (RSC Publishing))

Thank you for recommending this reference. We added it.

Line 496 - some actinobacteria. It doesn't get close to Streptomyces or Amycolatopsis.

We agree that it will be overstated without "some" and added it to the sentence.

Figure 1 - Very nice figure! Readability of the colors is very poor. All of the dark colored nodes in (b) look black. Also for color blind readers it is very difficult (speaking personally). You could use iWantHue (medialab.github.io) as a resource for making colored figures.

Thank you very much and apologize the poor color readability. We changed the colors in Fig 1b and additionally used different node types. This helped to improve the readability.

Figure 3a. Axis and phylogenetic tree distances are impossible to interpret. Are we looking m/z on the x axis of the barcoding?

We changed the size of the axis identifiers and the phylogenetic tree distances to improve the readability.

Figure 4. I'm fine with the R/S stereochemistry but I think labelling these with D/L nomenclature makes it more usable for the broad community outside of chemists. Also makes it easier to tie to the biosynthetic genes.

Thank you for this suggestion. We included the D/L nomenclature into Fig. 4.

Line XXX - You have discovered novel chemistry but I think you are over selling the work you have done here. CLPs would be considered 'low hanging fruit' since biosynthetically they are so easy to predict.

We agree with the reviewer and the adaptations made with this revision of the manuscript reduce the general hyperbole.

Reviewer #2 (Comments for the Author):

The overall approach seems straightforward and solid. However, I do have a few moderate to minor issues:

MODERATE ISSUES:

Lines 137 & 543 Please define what exactly is meant with "correctly annotated". Did this involve manual curation of the annotated datasets? In that case please state how you would recognize incorrectly annotated datasets

We agree that the word "correctly" is misleading in correlation with "annotated genomes". We did not manually curate the annotated datasets. Therefore, the word "correctly" was deleted in both sentences.

Lines 148-150 & lines 171-173: In lines 148-150 the authors first state that some classes display less significance for NP discovery than others, due to the lower number of BGC detected in the corresponding genomes. However in lines 171-173 they state that the simple number of BGCs detected is less important than their novelty (or "divergence" as the authors put it), a statement to which I whole-heartedly agree, but which strongly contradicts the first point. Wouldn't, for example, a taxon that yields a relatively low number but nonetheless high diversity of BGCs that are unique for this taxon be of actually more significance for novel NP discovery than taxa rich in BGCs that show close homologs in various other taxa? This should be better reflected in the wording.

That is very much true. Though, for the Bacteroidetes we are in a still very luxurious situation. We identify taxa that combine a high number of BGCs with a high level of novelty. We consider these taxa the best guess to identify many new metabolites. So in consequence we do believe, that these taxa (as e.g., Chitinophaga) have today the best "trade-off" and thus the highest significance for novel NP discovery. This correlation of amount and novelty was already stated in the manuscript (lines 225-235) as the main criteria why to focus for the cultivation approach on Chitinophaga.

During revision some sentences in the respective paragraphs were rephrased to improve readability.

Line 187: I think "other PKS" should also be placed in quotation marks

For clarification, we adjusted the sentence as following: "Nine of them belonged to the BiG-SCAPE BGC classes of other PKS ("PKSother"), "Terpenes" or "Other".

Lines 220-223: It would appear to me, that any sequence based screening for such broad functional categories as "secondary metabolite production" is likely to yield also some spurious and/or weakly supported results. Given also the large number of hypothetical proteins that are to be expected in the respective pan-genome of each phylum, many of such hits will be without any useful functional annotation and would therefore not appear in the MiBIG database, even if close homologs could be found in different phyla. Therefore, even if it might be true, I don't think the here presented analyses here actually prove a "tremendous uniqueness" of biosynthetic gene clusters within the Bacteroidetes. The authors should consider this and comment on it within the manuscript.

We agree with the reviewer as this is an interesting point of view. To determine the uniqueness one needs to include genomes from all phyla in the BiG-SCAPE analysis. If none of them (known BGCs and unknown BGCs) clusters with Bacteroidetes BGCs the uniqueness would be proven. Therefore, we adapted the wording accordingly to our type of analysis: "With >200 GCFs identified and only 12 of them annotated towards known BGCs and their metabolites, the sequential and compositional similarity analysis revealed a tremendous BGC diversity and uniqueness within the Bacteroidetes phylum, differing from the composition of known BGCs deposited in the MiBIG database."

Lines 542-544: Please define "WGS projects of strains of interest". Clearly this means non-circularised draft genomes, but since there is a huge number available, was that selection based on exactly (e.g. specific N50/N80-cutoffs?)

The selection was based on WGS projects that did contain less than 100 scaffolds. Hence, to mention this cutoff, we changed the text to: "...WGS projects of strains of interest (<100 scaffolds)."

Line 544: Supplementary table S8 does not seem to be available?

We are sorry if this was not available. It should have been uploaded with all other files in the initial submission. Supplementary table S8 lists all used 600 genomes with additional data, such as strain information, accession number, genome size, and total BGC count. We will upload this table during resubmission of the revised manuscript.

Line 288: "This high level of strain-specific metabolites": Could you provide some reference numbers here, in order for the reader to evaluate if this is actually a high number compared to other taxa?

We are sorry, but we are not aware of similar bucket numbers and analyses that would help to compare our numbers with peer-reviewed dataset from other taxa. The "high level" is related to the 1,154 strain-specific buckets which represents ~42% of all detected buckets (2,736) within our dataset.

Line 297-302: considering that the number of representative metabolites determined for the chitinophaga set alone (2736) is already significantly higher than the entire in-house reference database (1700), how representative is this comparison? Is it even possible to represent an equivalent range of metabolites in this database (especially considering the authors also show earlier that the range of metabolites captured is strongly dependent on the exact growth conditions)?

Thank you for this very interesting questions. We consider our in-house database in this context as a probe. It contains publicly known (~90%) and unknown (~10%) metabolites isolated from classical producer taxa (Fungi, Actinobacteria, Bacillus, Myxobacteria but also Cyanobacteria). Its comparison to MS-data (from crude extracts) from these classical producer taxa cultivated under various cultivation conditions identifies known as well as still many unknown buckets. The convincing observation comparing this database with the Bacteroidetes extracts is thus not the level/amount of unknowns but the lack of detection of any known metabolite. This finding was also confirmed by the comparison of the LC-MS/MS data to an even bigger database - Antibase (>40k NPs). The comparison to the Antibase data is now mentioned following the sentence that refers to our internal database.

Lines 411-414: Is data shown for this anywhere in the manuscript or tal ? please reference.

Thank you for pointing this out. We have added the reference Fig. S37 for the LC-MS confirmation of the iron complexed form and reference Table S4 for the bioactivity of the iron complexed chitinopeptin A.

MINOR ISSUES:

Lines 108 and 110-112: first mentioning that the NP production is limited to a few taxonomic hotspots within the Bacteroidetes, rather than the whole phylum but then announcing an "overall high potential to discover novel scaffolds from this phylum" may easily appear contradictory. The authors are actually distinguishing between the sheer NUMBER of potential NP biosynthesis gene clusters concentrated mostly in a few Bacteroidetes taxa and the apparent NOVELTY of said clusters, which appears to be relatively high for the entire Bacteroidetes phylum. In order to avoid misunderstanding, the authors may want to rephrase that section slightly in order to make that more clear.

We rephrased this section slightly in order to clearly state the combination of sheer NUMBER and apparent NOVELTY as driving motivation for data rating and eventual strain selection for the metabolomics studies.

Lines 128-129: "The large and diverse phylum Bacteroidetes harbors Gram-stain-negative, chemo-organotrophic, 128 non-spore forming rod shaped bacteria, graded into six so-called classes. " I think some literature should be cited also for these general properties of the phylum.

Reference "Paster, B. J., Dewhirst, F. E., Olsen, I., and Fraser, G. J. (1994). Phylogeny of Bacteroides, Prevotella, and Porphyromonas spp. and related bacteria. J. Bacteriol. 176, 725–732." has been added for the general properties. Additionally, references 48 and 49 have been added to this sentence for the taxonomic classification of the Bacteroidetes phylum into 6 classes. Both references had already been used in a later sentence.

Line 141-143. This kind of figure description belongs into the respective figure legend, not into the main text.

We updated the Lines 141-143 accordingly.

Line 185. I think a more precise wording would be "...only 12 GCFs clustered with MiBIG reference BGCs of known function"?

Thank you for this precise wording. We have changed it accordingly.

Line 401: "3018 medium" does not seem to be commonly familiar? please provide a reference.

The reviewer is right, it is an internal identifier for a medium used in-house and not a commonly familiar one. The composition is written in the Materials and Methods section. For clarification, we adjusted the sentence as follows: "To investigate the impact of iron on the CLPs production, *C. eiseniae* was cultured in 3018 medium (for composition see Materials and Methods section) supplemented with different iron concentrations."

Line 404: why directly specify the exact number of fungi, but not of gram positive and gram negative bacteria here? It would appear better to specify "[...] tested against 8 gram-negative and 5 gram-positive bacterial strains, as well as [...]"

We agree and specified the exact numbers.

Line 511: "[...] is in similar range than [...]". Wording seems strange. Maybe change to "shows a similar range compared to [...]"?

Thank you for this suggestion. We agree and changed it accordingly.

ADDITIONAL NOTES ON WORDING:

Abstract line 39: perhaps drop the "the" in "we isolated THE new iron chelating[...]"?

We do not like to drop the "the", as the chitinopeptins are named the first time later in this sentence. It does not sound correct without the "the". Additionally, we change the word "Sourcing" into "Mining" because reviewer 1 suggested using this word. Sentence: "~~Sourcing~~Mining this dataset, we isolated the new iron chelating nonribosomally synthesized cyclic tetradeca- and pentadecalipodepsipeptide antibiotics chitinopeptins with activity against *Candida*,... ."

Lines 95&96: "[...] towards a to date yet underexplored [...]" wording seems a bit off. Maybe change to "[...] toward a currently still underexplored [...]"?

We agree and have changed it throughout the whole manuscript.

Line 144-145: "[...]enabled classes and genera comparison in terms of [...]" seems like strange wording. Maybe "[...] enabled comparisons between different classes and genera in terms of [...]" would be more correct?

Thank you, we have adjusted it accordingly.

Line 168: The term "second best genus" seems oddly subjective... Maybe a better wording would be "The genus with the second highest BGC load within the Bacteroidetes phylum was *Taibaiella* [...]"?

We agree and have adjusted it accordingly.

Line 277: "..were detected after seven then after four ..." à correct "then" to "than"

We have corrected it.

Line 288: over the course of the manuscript it can be a little hard to follow which reference sets were used for which analyses, therefore the wording "[...] 10 out of 25 [...]" may seem a little confusing. It may be better to specify here again: "[...] 10 out of the 25 total analyzed strains [...]"

We agree and have adjusted it accordingly.

Sincerely yours,
On behalf of the authors,

Till Schäberle

March 29, 2022

Prof. Till F. Schäberle
University of Giessen
Institute for Insect Biotechnology
Ohlebergsweg 12
Giessen 35392
Germany

Re: Spectrum02479-21R1 (Genomic and chemical decryption of the Bacteroidetes phylum for its potential to biosynthesize natural products)

Dear Prof. Till F. Schäberle:

Your manuscript has been accepted, and I am forwarding it to the ASM Journals Department for publication. You will be notified when your proofs are ready to be viewed.

Sincerely,

Eva Sonnenschein
Editor, Microbiology Spectrum

Journals Department
Supplemental file 3: Accept
Supplemental Material: Accept